# Superscattering emerging from the physics of bound states in the continuum

Adrià Canós Valero ®[1,2] ✉, Hadi K. Shamkhi[2,3], Anton S. Kupriianov[4], Thomas Weiss ®[1], Alexander A. Pavlov[5], Dmitrii Redka ®[6], Vjaceslavs Bobrovs ®[7], Yuri Kivshar ®[8] ✉ & Alexander S. Shalin ®[9,10] ✉

We study the Mie-like scattering from an open subwavelength resonator made of a high-index dielectric material, when its parameters are tuned to the regime of interfering resonances. We uncover a novel mechanism of superscattering, closely linked to strong coupling of the resonant modes and described by the physics of bound states in the continuum (BICs). We demonstrate that the enhanced scattering occurs due to constructive interference described by the Friedrich-Wintgen mechanism of interfering resonances, allowing to push the scattering cross section of a multipole resonance beyond the currently established limit. We develop a general non-Hermitian model to describe interfering resonances of the quasi-normal modes, and study subwavelength dielectric nonspherical resonators exhibiting avoided crossing resonances associated with quasi-BIC states. We confirm our theoretical findings by a scattering experiment conducted in the microwave frequency range. Our results reveal a new strategy to boost scattering from non-Hermitian systems, suggesting important implications for metadevices.

Non-Hermitian physics offers a wide range of unusual phenomena not accessible for purely Hermitian systems[1]. In recent years, there has been tremendous progress in the implementations of non-Hermitian platforms in optics, with discoveries of many intriguing effects that may occur in lossy or gain-compensated optical structures. Being motivated by the studies of parity-time ($\mathcal{PT}$) -symmetric systems, a novel field of non-Hermitian photonics emerged[1], taking advantage of new degrees of freedom offered by complex energy landscapes[2–6]. The advancements are particularly exciting for subwavelength photonics, allowing to study of unconventional regimes of light-matter interaction such as exceptional points[7,8] and dark states[9–11].

Importantly, the eigenvalues of an isolated optical resonator with uncompensated radiative losses are always complex.

Nevertheless, they can be controlled by engineering the resonator parameters to achieve the regime of bound states in the continuum (BIC) with ultrahigh quality factors ($Q$-factors) and strong energy localization[12]. This regime arises due to the destructive interference within the modes of the same radiation channel, as a consequence of the Friedrich-Wintgen (FW) mechanism of interfering resonances[13]. While a bimodal system coupled to one channel of the continuum is well understood, the subtleties underlying multiple-channel inter-actions are yet to be exploited in photonics. Destructive interference leads to a quasi-BIC regime and the suppression of radiation in one channel. Here, we pose the question, of whether constructive inter-ference of a quasi-BIC state and a low-$Q$ mode in a multi-channel structure can boost radiation beyond the limit for an isotropic

[1]Institute of Physics, University of Graz, and NAWI Graz, 8010 Graz, Austria. [2]ITMO University, St. Petersburg 197101, Russia. [3]Institute of Materials Research and Engineering, Agency for Science, Technology and Research, 2 Fusionopolis Way, Innovis #08-03, Singapore 138634, Republic of Singapore. [4]College of Physics, Jilin University, Changchun 130012, China. [5]Institute of Nanotechnology of Microelectronics, Moscow 119991, Russia. [6]Electrotechnical University LETI, St. Petersburg 197376, Russia. [7]Riga Technical University, Institute of Telecommunications, Riga 1048, Latvia. [8]Nonlinear Physics Centre, Research School of Physics, Australian National University, Canberra, ACT 2601, Australia. [9]Center for Photonics and 2D Materials, Moscow Institute of Physics and Technology, Dolgoprudny 141700, Russia. [10]MSU, Faculty of Physics, Moscow 119991, Russia. ✉e-mail: adria.canos5@gmail.com; yuri.kivshar@anu.edu.au; alexandesh@gmail.com

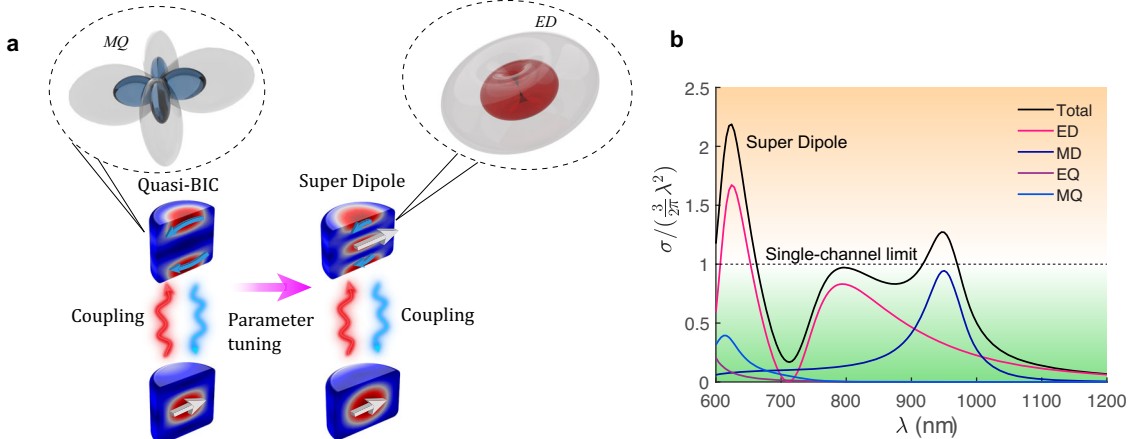

**Fig. 1 | Superscattering from the physics of BICs. a** Concept of BIC-inspired superscattering in an isolated resonator. Strong coupling of two modes reshapes both their near fields and scattering patterns as a function of a tuning parameter. The interfering resonances lead to a quasi-BIC state (destructive interference) and induce power redistribution between multipolar scattering channels leading to super-dipole radiation (constructive interference). **b** Super-dipole resonance arising in the scattering cross-section of a dielectric cylinder with refractive index $n_p \sim 3.8$, radius 130 nm, and height 180 nm. The scattering cross-section of the electric dipole channel significantly exceeds the single-channel limit. This is in contrast with conventional superscattering, where several multipole resonances need to be overlapped.

scatterer, realizing superscattering[14,15]. Until now, superscattering was known to originate only from a degeneracy of multipolar resonances[14–22], which, when spectrally overlapped, exceeded the single-channel cross-section.

In this work, we demonstrate that strong coupling between two modes can lead to a previously unknown regime of superscattering in subwavelength resonators, in addition to the quasi-BIC states (see Fig. 1a). We reveal that mode coupling induces power redistribution between two scattering channels, allowing to overcome the single-channel scattering limit and control not only the $Q$-factor, but also enhance the power scattered by a multipole, (e.g., the electric dipole) beyond the limit, as demonstrated in Fig. 1b. Unlike the recent proposal in[23], there is no need to introduce gain in the resonator. We first formulate a general phenomenological model and later employ rigorous perturbation theory for quasi-normal modes (QNMs) of non-Hermitian resonators[24,25] to design several examples of subwavelength cavities with broken spherical symmetry.

## Results

### Enhancing scattering by finite objects

We start our analysis by overviewing how superscattering arises through the mechanism originally proposed in[16]. Consider a particle possessing either spherical symmetry, or dimensions much smaller than the incident wavelength, illuminated along the $z$ axis by an incident plane wave. In all that follows, we consider its center of mass is taken as the origin of the coordinate system.

In or outside the smallest spherical region surrounding such particle, the electromagnetic fields can be expressed as a combination of multipolar waves, i.e., $\boldsymbol{E}(r) = \sum_\tau c_\tau \boldsymbol{W}_\tau(r)$. The triplet $(\tau) \equiv (\ell, m, p)$ represents a scattering 'channel' $\tau$, through which the particle can exchange power with the environment. The first number indicates the total angular momentum, so that $l = 1$ is a dipole, $l = 2$ a quadrupole, and so forth. The second is the absolute value of the projection of angular momentum to the z-axis, while $p$ denotes the magnetic or electric character of the multipole ($p = 1$ for electric or $p = 2$ magnetic). Each multipolar wave can be decomposed further into an outgoing (−) and an incoming (+) wave, so that the field in one channel is alternatively expressed as $\boldsymbol{E}_\tau(r) = s_\tau^+ \boldsymbol{W}_\tau^+(r) + s_\tau^- \boldsymbol{W}_\tau^-(r)$. $s_\tau^{+(-)}$ are the incoming (outgoing) coefficients in channel $\tau$. With a suitable normalization, $|s_\tau^{+(-)}|^2$ corresponds to the power carried towards or away from the particle in every multipole channel.

The effect of the scatterer is completely described by the 'reflection' coefficients $R_\tau \equiv s_\tau^- / s_\tau^+$. Furthermore, energy conservation dictates $|R_\tau| \leq 1$ for passive scatterers. The scattering cross section for each channel is then given by

$$\sigma_\tau = \frac{2\ell + 1}{8\pi}\lambda^2 |1 - R_\tau|^2 \qquad (1)$$

The limit is attained for $R_\tau = -1$, and yields $\sigma_{\text{Max}} = (2\ell + 1)\lambda^2/2\pi$. For example, in the case of a dipole (electric or magnetic), the limit is $\sigma_{\text{Max}} = 3\lambda^2/2\pi$. In a scatterer with negligible absorption losses, this limit can be achieved at a multipolar resonance[16]. To each resonance, one can associate an underlying quasinormal mode (QNM), with a complex eigenfrequency $\tilde{\omega}_m = \omega_m - i\gamma_m$. For small $\gamma_m$, a multipolar resonance appears in the real frequency spectrum, centered around $\omega_m$ and having a linewidth of $2\gamma_m$.

To design a superscatterer, the resonant frequencies of several QNMs associated to different multipolar channels must be brought together by a smart design of the particle geometry, so that the total scattering cross section, given by the sum of their contributions, exceeds the limit. This is typically done so by adding material layers of different thickness to a sphere or an elongated rod[14,21,26]. The superscattering regime was very recently experimentally verified for the first time[14].

What happens in the absence of spherical symmetry? Intriguingly, it was shown that larger bounds on total extinction could be attained for lossy nonspherical shapes[27], even for deeply subwavelength plasmonic particles. The enhancement, however, was mostly delivered by absorption from such particles, and still remained significantly below the single-channel limit (for a detailed comparison, we refer the reader to the Supplementary Information S12 and Figure S6). Finite plasmonic nanorods were also numerically investigated[22], exhibiting enhanced cross-sections for some well-chosen geometrical parameters (Figure S6). Despite these works, focused in plasmonic cavities with large absorption cross-sections, there appeared to be no qualitative differences between a spherical shape and the general case.

In fact, scattering by nonspherical objects, even with subwavelength dimensions, is described by a matrix R with potentially nonzero off-diagonal components of the form $R_{\tau\tau'} \equiv s_\tau^- / s_{\tau'}^+$. They relate the incoming wave in channel $\tau'$, to the outgoing wave in channel $\tau$. This has an important consequence: since the limit in Eq. (1) relies on

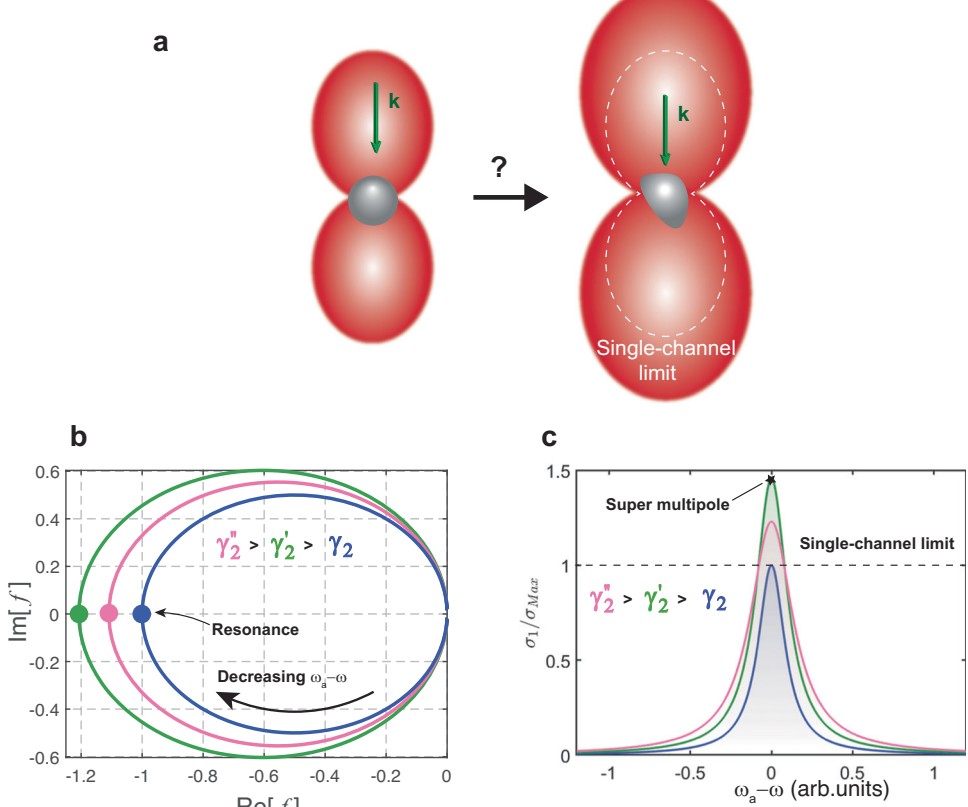

**Fig. 2 | Boosting scattering of a multipolar resonance. a** Artistic picture illustrating the question at hand: can symmetry break enhance the scattering cross-section of a multipole beyond the limit? Red lobes represent the scattering pattern characteristic of an electric or magnetic dipole. On the left-hand side, a spherical scatterer is illuminated by a plane wave with momentum $k$. On the right-hand side, a nonspherical scatterer displays a similar dipolar scattering pattern, but significantly enhanced. **b, c** General model of a single QNM $|a\rangle$ compatible with two scattering channels $\tau = 1, 2$. **b** Shows the evolution of the real and imaginary parts of $f(\omega)$, whose modulus determines the scattering enhancement (see details in text), as a

function of the detuning $\omega_a - \omega$, for fixed $\gamma_1 = 1$ (radiation rate to channel 1 in normalized units) and increasing $\gamma_2$ (radiation rate to channel 2). Dark blue: $\gamma_2 = 0$, green: $\gamma_2' = 0.017$, pink: $\gamma_2'' = 0.06$. Resonance takes place when $\omega_a - \omega = 0$. A maximum of $f(\omega)$ occurs for a critical value of radiation rate to the second channel (green curve), and then progressively decreases. **c** Scattering cross section to channel 1 for the same cases studied in **b**, normalized by the single-channel limit, denoted as $\sigma_{Max}$. When $\gamma_2' = 0.017$, the scattering cross section of the channel significantly exceeds the limit, leading to a 'super' multipole resonance.

the $R$-matrix being diagonal, (i.e., there is no power exchange between the multipole channels), the latter, in principle, ceases to be valid.

We realize that, in general, depending on the symmetry of the object, the scattering cross section of one multipole can receive contributions from other multipoles. Intriguingly, this suggests that the strength of a multipole could, in theory, be boosted beyond the conventionally accepted limit, as depicted in Fig. 2a. In stark contrast with conventional superscattering, by being able to enhance the cross section of a single multipole, we could not only enhance overall scattering, but also manipulate the radiation pattern, for instance making it larger without significantly altering its shape, as illustrated in Fig. 2a. So far, this effect has proven to be elusive, and has not been reported in the literature.

**Super-multipole resonances**

To verify this possibility, we assume only two scattering channels (multipoles) $\tau = 1, 2$ are important, and, after some assumptions we derive an ad-hoc expression for the scattering cross section of channel 1 (derivation provided in the Supplementary Information S3):

$$\sigma_1/\sigma_{Max} = \frac{1}{4}|1 - R_{11} - R_{12}|^2 \equiv |f(\omega)|^2 \qquad (2)$$

The function $f(\omega)$ can be complex, and its modulus will determine the ultimate limit for scattering in this case. Namely, if $|f(\omega)|>1$, the

single-channel limit could be exceeded, since $\sigma_1$ would then be larger than $\sigma_{Max}$. To derive an analytical expression for it, we investigate a hypothetical structure supporting a single QNM $|a\rangle$ which, due to an a priori unknown mechanism, is coupled to the two scattering channels. The radiation rate of is the sum of radiation rates to the two channels, i.e., $\gamma_a = \gamma_1 + \gamma_2$. According to temporal coupled mode theory[18,28] (TCMT), we get $f(\omega) = i\frac{\sqrt{\gamma_1\gamma_2} + \gamma_1}{\omega_a - \omega - i(\gamma_1 + \gamma_2)}$ (refer to Supplementary Information S4 for details).

What happens when the radiation rate to the second channel increases? Fixing $\gamma_1 = 1$ (in normalized units), we plot $f(\omega)$ as a function of detuning from resonance $\omega_a - \omega$, for different values of $\gamma_2$ (Fig. 2b). In all cases, the maximum occurs at resonance (zero detuning). If $\gamma_2 = 0$, $|f(\omega)| = 1$, since the QNM can only radiate to one channel. This is the conventional case. Strikingly, there exists a critical $\gamma_2$ for which $|f(\omega)|$ reaches a maximum exceeding 1 (green curve in Fig. 2b). Interestingly, for radiation rates larger than the critical (pink curve in Fig. 2b) we observe a progressive degradation of the enhancement, confirming that there indeed exists an optimal, small $\gamma_2$ where the cross section is maximized beyond the limit.

To provide more insight, the scattering cross section of channel 1 is displayed in Fig. 2c, for the different cases shown in Fig. 2b. It can be clearly seen that the green spectrum exceeds the limit by almost 1.5 times. It should be noted that this is the case despite a clear broadening of the resonance due to the additional losses. Thus, contrary to what is

widely accepted, radiation losses can contribute to an enhancement of the channel cross section, instead of degrading it. We refer to this novel regime as a 'super-multipole', in contrast with conventional superscattering, where several QNMs need to be overlapped in the spectrum.

With this first model, we have predicted the existence of super-multipole resonances capable of enhancing the cross section of one channel beyond the limit. Symmetry breaking is a necessary but not sufficient condition for their formation. In particular, a careful control of the radiation rate to each of the multipoles involved is required for their realization. It is not evident how this control can be achieved in practice.

In the following section, we show that, in the vicinity of a quasi-BIC, symmetry breaking has a strong impact on the scattering cross section. QNMs belonging to different multipoles can couple strongly in the near field, leading to an avoided crossing and acquiring a mixed multipolar character, which allows to easily modify their radiation rates. Then, defying conventional intuition, a single super-multipole resonance can drive the scattering cross section of a multipole beyond the single-channel limit (as well as the total scattering cross section). Super-multipoles exist as a natural counterpart of a quasi-BIC, where the contribution of a QNM to one or more scattering channels is forbidden.

## Super-multipoles emerging from quasi-BICs

We now extend our TCMT model above to the case of a structure supporting two QNMs $|a\rangle$, $|b\rangle$, each compatible, respectively, with a single scattering channel $\tau = 1, 2$. These can be, for instance, two multipolar QNMs of a suitably designed spherical scatterer (e.g., the electric dipole and magnetic quadrupole modes). In a real system, there might also be contributions from non-resonant QNMs forming a 'background', as shown later. These are, however, neglected in our preliminary analysis. The model is used solely for illustration purposes. Later, the results are verified with rigorous cavity perturbation theory in a realistic nanoresonator, as well as with microwave experiments.

After breaking the spherical symmetry in some fashion, the two QNMs can couple in the near field, leading to an 'effective' Hamiltonian of the form (refer to the Supplementary Information S2):

$$\mathcal{H}_0(\zeta) = \begin{pmatrix} \widetilde{\omega}_a(\zeta) & \kappa(\zeta) \\ \kappa(\zeta) & \widetilde{\omega}_b(\zeta) \end{pmatrix} \tag{3}$$

Diagonalizing $\mathcal{H}_0(\zeta)$ results in two new hybrid QNMs $|u\rangle$, $|d\rangle$ that are a combination of the original ones. Thus, the new modes have a mixed multipolar nature. Accidentally, (or due to symmetry), the coupling coefficient can vanish, and the hybrid QNMs become $|u\rangle = |a\rangle$ and $|d\rangle = |b\rangle$. We consider that, in general, the uncoupled eigenfrequencies and the coupling coefficient (here assumed to be real), are a function of a geometrical parameter $\zeta$. In our first example in the next section, $\zeta$ will be related to the ellipticity of the particle.

In general, Eq. (3) does not only describe the hybridization of the QNMs of a sphere, but that of any particle without spherical symmetry, such as a finite cylinder, as shown later. If we assume that channel 1 is a low order multipole (for instance a dipole), while channel 2 is a higher order one, (for instance a quadrupole), radiative losses in channel 2 are significantly lower. Then, for $\kappa = 0$, QNM $|b\rangle$ corresponds to a quasi-BIC with high $Q$-factor. This is because $|b\rangle$ is, by assumption in our model, completely unmatched from the lowest order multipoles[12,29].

Assuming time-harmonic dependence of all fields in the form $e^{-i\omega t}$, we can derive an expression for the $R$-matrix[28,30], (in the remainder of this work, unless written explicitly, we omit the $\zeta$ dependence for the sake of brevity):

$$\mathcal{R}(\omega) = \mathbb{I}_2 + 2\mathcal{T}(\omega) \tag{4}$$

$$\mathcal{T}(\omega) = iD\left[\mathcal{H}_0(\zeta) - \mathbb{I}_2\omega\right]^{-1}D^T \tag{5}$$

where $\mathbb{I}_2$ is the $2 \times 2$ identity matrix, and $D$ is a matrix connecting the QNMs to the multipole fields[31]. In the absence of any scatterer, $\mathbb{I}_2$ guarantees the incoming waves are perfectly reflected to the outgoing ones.

In this first scenario, $D = \mathrm{diag}(d_1, d_2)$. This ensures that the original QNMs are matched to different multipolar channels. In particular, $|a\rangle$ is matched only with multipole 1, and $|b\rangle$ with 2. The radiative losses of each uncoupled mode are given by $\gamma_{a,b} = d_{1,2}^2$. Emulating a realistic situation (as shown in the next section), we consider channel 1 corresponds to a dipole (electric or magnetic), and channel 2 to a quadrupole (magnetic or electric). Since dipolar QNMs leak strongly to the far field, $d_1 \gg d_2$, hence it follows that $\gamma_a \gg \gamma_b$. Due to this, as mentioned earlier, when $\kappa = 0$, QNM $|d\rangle = |b\rangle$ and displays a peak in its $Q$-factor, which corresponds to a quasi-BIC.

Based on the model [Eq. (4)], we derive an expression for the scattering cross section of the dipole channel (channel 1), normalized by its conventional limit, $\sigma_{Max} = 3\lambda^2/2\pi$:

$$\sigma_1/\sigma_{Max} = |f(\omega)|^2 \tag{6}$$

$$f(\omega) = \frac{id_1^2(\widetilde{\omega}_u - \widetilde{\omega}_0)}{(\widetilde{\omega}_u - \widetilde{\omega}_d)(\omega - \widetilde{\omega}_u)} - \frac{id_1^2(\widetilde{\omega}_d - \widetilde{\omega}_0)}{(\widetilde{\omega}_u - \widetilde{\omega}_d)(\omega - \widetilde{\omega}_d)} \tag{7}$$

For details, see the Supplementary Information S4. Each of the two terms in Eq. (7) accounts for the contribution of a hybrid QNM to the channel cross section. On the other hand, $\widetilde{\omega}_0 = \widetilde{\omega}_b - \kappa d_2/d_1$ is a zero of the cross section. Near the resonance frequencies of the hybrid QNMs, only one of the terms in Eq. (7) is dominant. The mechanism through which the single-channel limit can be exceeded is illustrated in Fig. 3.

In the absence of coupling ($\kappa = 0$), $\widetilde{\omega}_u = \widetilde{\omega}_a$, $\widetilde{\omega}_d = \widetilde{\omega}_b$ and $\widetilde{\omega}_0 = \widetilde{\omega}_b$, so that the contribution of QNM $|d\rangle$ in Eq. (7) vanishes, as shown in Fig. 3a. We provide a scheme of this situation in Fig. 3b. The pole of $f(\omega)$ associated with $|d\rangle$ annihilates with the zero, cancelling its contribution to the cross section. In other words, QNM $|d\rangle$ leaks only through the quadrupole channel and exhibits a high $Q$-factor (a quasi-BIC). Then, the maxima in the cross section for the dipole channel occurs only at $\mathrm{Re}(\widetilde{\omega}_u) = \omega_a$ and is bounded to $\sigma_{Max}$, as in the usual case.

Introducing coupling ($\kappa \neq 0$), results in several interesting effects, as shown in Fig. 3c, d. Firstly, in Fig. 3d, the pole associated with $|d\rangle$ and the zero of $f(\omega)$ are shifted away from one another. Thus, both the zero and the new pole contribute to the dipole channel (Fig. 3c). The dipole channel is now 'open' for QNM $|d\rangle$. Consequently, its radiative losses are increased (i.e., lower $Q$-factor), and the pole is pushed deeper into the complex plane (see Fig. 3d). Remarkably, we notice that at $\mathrm{Re}(\widetilde{\omega}_u)$ or $\mathrm{Re}(\widetilde{\omega}_d)$, the maxima in the cross section of the channel is no longer bounded to $\sigma_{Max}$ in Eq. (6). This is also true if one of the terms in Eq. (7) is dominant. Thus, by inducing coupling between two multipole resonances, we can create a hybrid mode that is able to enhance scattering by itself beyond the established limit.

Such peculiar behavior can be observed in Fig. 3e, where we plotted the normalized dipole cross section as a function of the tuning parameter $\zeta$. The functional form of the different elements in $\mathcal{H}_0(\zeta)$ is given in the caption of Fig. 3. They are justified by first order QNM perturbation theory[25,32,33] (see Supplementary Information S2). In particular, $\kappa(\zeta) = a_3\zeta$, where $a_3$ is a constant. Due to the nonzero coupling for any $\zeta \neq 0$, the hybrid QNMs 'avoid' the crossing due to strong interaction between the original modes (i.e., strong coupling). This is a particular example of the Friedrich-Wintgen mechanism[13], which has been shown to lead to true BICs in extended structures[34], or extremely high-$Q$ quasi-BICs in isolated cavities[12,29,35]. For the case under

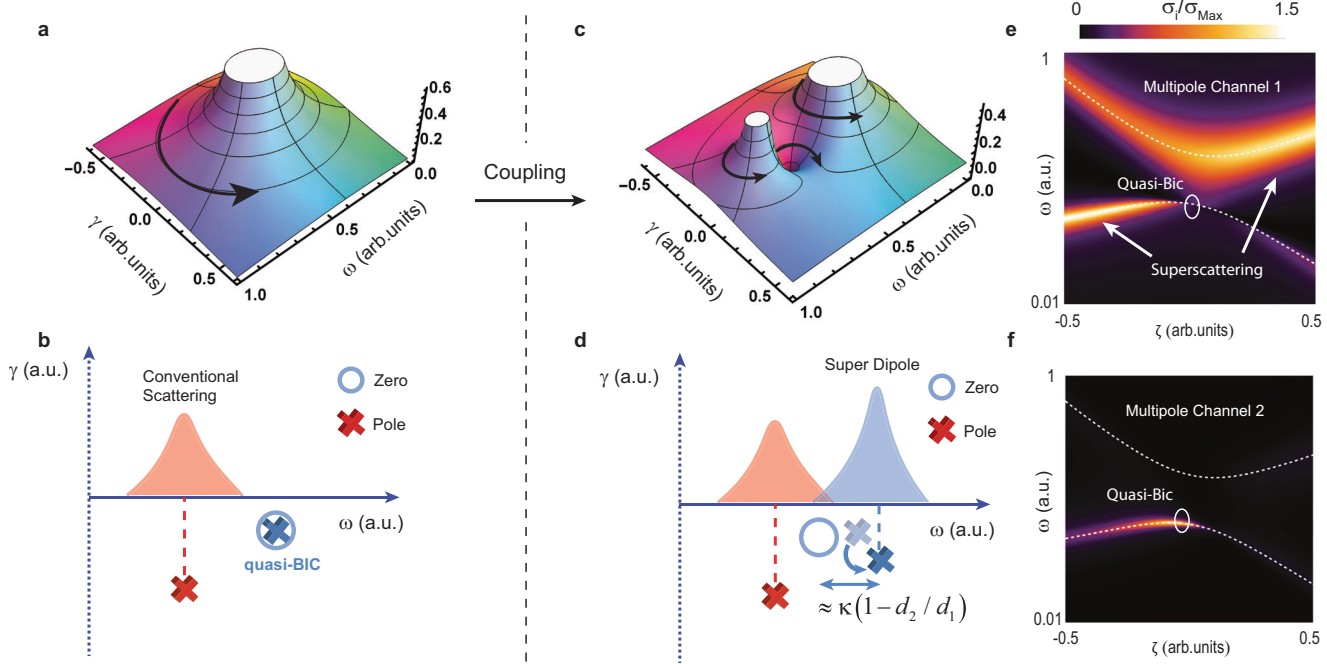

**Fig. 3 | Toy model describing the physics of the new superscattering regime.**
**a** Complex plot of Eq. (7) for $\kappa = 0$ (spherical symmetry, no coupling). Clockwise black arrows denote zeros of the function, while counterclockwise ones denote poles (resonances). In this first case, only QNM $|u\rangle = |a\rangle$ contributes since the pole induced by QNM $|d\rangle$ annihilates with the zero. **b** Schematic illustration of the zero-pole annihilation, corresponding to a quasi-BIC, and conventional multipole scattering. **c, d** Same as in **a, b**, but for the case with $\kappa \neq 0$. Now, the coupling separates in the complex plane the pole associated with $|u\rangle$ and the zero. In exchange, the QNM gains additional losses, and is allowed to contribute to scattering, which is no longer bounded by the limit [blue-shaded area in **d**]. **c**–**f** scattering cross section for

the dipole channel 1, and the quadrupole channel 2, respectively. All the values have been normalized by the limit of the dipole cross section $\sigma_{Max} = 3\lambda^2/2\pi$. For illustration purposes, we consider $\omega_a(\zeta) = \omega_a^{(0)}/(1 + a_1\zeta), \omega_b(\zeta) = \omega_b^{(0)}/(1 + a_2\zeta), \kappa(\zeta) = a_3$, with $\omega_a^{(0)} = 0.6, \omega_b^{(0)} = 0.4$, $a_1 = 0.4, a_2 = 0.38$, $a_3 = 0.5$. White dashed lines indicate the path followed by the hybrid eigenfrequencies $\tilde{\omega}_{u,d}$. At the quasi-BIC, scattering to the dipole channel by QNM $|d\rangle$ is completely suppressed. Once coupling 'opens' the channel to $|d\rangle$, the scattering energy is transferred from the quadrupole channel to the dipole channel, exceeding its limit and reaching the 'super dipole' regime.

consideration, the quasi-BIC appears in the lower branch at $\zeta = 0$, since $\kappa(0) = 0$ and the contribution of QNM $|d\rangle$ to the dipole vanishes, while a resonance arises in the quadrupole channel (Fig. 3f). Only the lossy QNM $|u\rangle$ is matched to the dipole. This corresponds to the usual scenario, where the dipole cross section cannot exceed the traditional limit.

For any $\zeta \neq 0$, there is coupling between the QNMs, and the lower branch can scatter as a dipole (Fig. 3e), which, from the mechanism discussed above, can largely exceed the limit in a sphere. We therefore refer to this unusual resonance as a 'super dipole'. We bear in mind that, due to the mixed multipolar nature of the resonance, there is a small contribution to the quadrupole channel. We also remark that QNM $|u\rangle$ can display similar features for positive $\zeta$, as shown in Fig. 3e.

Summarizing, we have derived a simple theory that allows us to describe the Friedrich-Wintgen mechanism leading to quasi-BICs in isolated cavities. We have demonstrated the occurrence of yet another surprising effect, namely, the possibility to create a resonance capable of enhancing scattering beyond the accepted limit in a subwavelength object, reaching the superscattering regime.

We stress again that our mechanism is in stark contrast with the conventional way to achieve superscattering. Typically, several orthogonal resonances (several QNMs matched to different multipole channels) must be overlapped at the same spectral position. Instead, here we exploit the Friedrich-Wintgen mechanism to 'open' the dipole channel to a quasi-BIC, forming a super dipole mode. So far, however, our predictions remain purely theoretical. In what follows, we employ multipolar theory and group-theoretical arguments to design two subwavelength scatterers displaying super dipole modes. We then

verify our results in the microwave range, confirming their existence for the first time.

## Subwavelength nanoresonators

We consider a Si nanosphere in air, illuminated with a normally incident, linearly polarized plane wave, with radius 100 nm. In the visible range, it supports two QNMs matched to the electric dipole (ED) and magnetic quadrupole (MQ) channels, respectively. Their electric field distributions are shown in the lower panel of Fig. 4c. We use the same notation as in the previous section, and label them as $|a\rangle$, $|b\rangle$. The far-field projections of the QNMs correspond to an $x$-polarized ED ($p_x$) and the $yz$ component of the MQ moment ($M_{yz}$). The reason for our choice of QNMs will become clear in the following. Note that the chosen scatterer is deeply subwavelength, with the radius being at least five times smaller than the incident wavelength.

In order to couple both QNMs, it is necessary to break the spherical symmetry in some fashion. A simple way to do so is by reducing the rotational symmetry in the plane parallel to the direction of propagation, (refer to schematic insets in Fig. 4b). Formally, the point group of the resonator changes from $O(3)$ (spherical symmetry) to $D_h$ (cylindrical symmetry). Then, multipolar modes with the same parity (as in the case for the chosen QNMs), can couple[36].

As depicted in Fig. 4b, we perform a controlled symmetry breaking by changing the ratio between the two orthogonal axes of the resonator $r_\parallel, r_\perp$. In this case $\zeta \equiv 1 - r_\perp/r_\parallel$. Cavity perturbation theory[25] (see Sec. S1 of the Supplementary Information), predicts the formation of hybrid QNMs, whose far fields are now a combination of $p_x$ and $M_{yz}$. This results in an avoided crossing in the eigenfrequency spectrum (Fig. 4a).

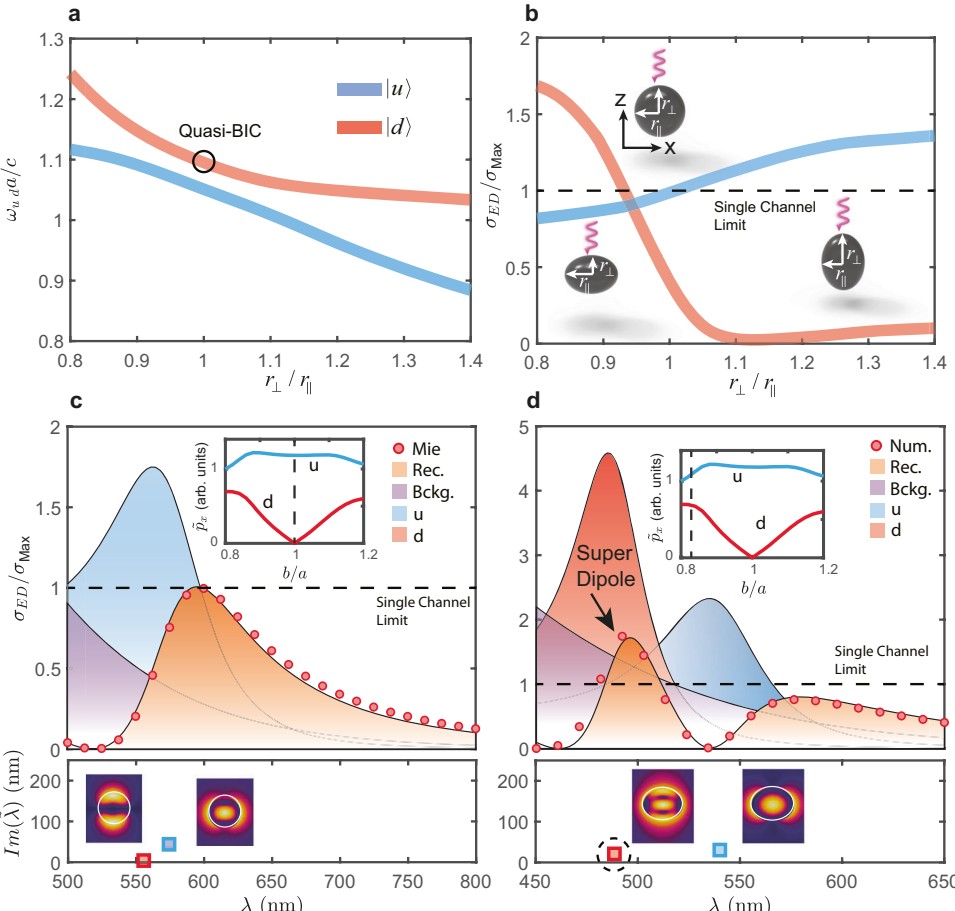

**Fig. 4 | Super-ED resonance in a dielectric nanoellipsoid. a** Evolution of the resonant frequencies of the ED and MQ modes of a silicon nanosphere (100 nm radius), when breaking the rotational symmetry along one of its axis, [generating an ellipsoid with semiaxis $r_\parallel$, $r_\perp$ indicated in the insets of **b**]. An avoided crossing, the hallmark of strong coupling, can be clearly observed. **b** Scattering cross-section of the ED channel at the two resonance maxima as a function of ellipticity, under normally incident, linearly polarized plane wave illumination, with momentum oriented along the axis with broken rotational symmetry. For $r_\perp/r_\parallel = 1$ (a sphere), the cross-section is bounded to 1, and QNM $|d\rangle$ features a quasi-BIC. For an oblate ellipsoid (left inset), QNM $|d\rangle$ becomes a super-ED, and vice versa for QNM $|u\rangle$ (right inset). **c**, **d** Contributions of the QNMs to the scattering cross-section of the ED channel in a sphere and a perturbed spheroid. **c** Sphere ($r_\perp/r_\parallel = 1$). Upper panel: ED scattering cross section obtained with conventional Mie theory and its reconstruction with QNMs (Rec.). The curves labeled *Bckg, u, d* are evaluated as $|\boldsymbol{p}_m|^2/I_0$

where $m =$ Bckg,$u$,$d$ correspond to the non-resonant term[52,53], the $|u\rangle$ and $|d\rangle$ QNMs, respectively. Inset: Normalized ED content of QNMs $|u\rangle$,$|d\rangle$, calculated as in ref. 54. They provide an estimation of the matching of a QNM to the ED channel. At the quasi-BIC there is no matching, thus $\widetilde{p}_x = 0$. Lower panel: eigen wavelengths of $|u\rangle$, $|d\rangle$, in the complex plane, defined as $\widetilde{\lambda}_m = 2\pi c/\widetilde{\omega}_m$, and their field distributions in the $x$–$z$ plane ($|\widetilde{E}_m|$, in arbitrary units). **d** Same as in **c**, but for an ellipsoid with $r_\perp/r_\parallel = 0.85$. Note that QNM $|d\rangle$ is now matched to the ED channel, since its ED content is nonzero (upper panel inset). Thus, it contributes to the cross section, and a super dipole appears in the spectra. The dashed circle in the lower panel indicates $\widetilde{\lambda}_d$, which has been pulled deeper in the complex plane due to the additional radiation losses, in accordance with the mechanism described in the previous section. Due to coupling with $|u\rangle$, the field distribution of $|d\rangle$ is drastically reshaped in comparison with the sphere. More details on the coupling mechanism can be found in Figure S1 of the Supplementary Information.

Now, an incoming ED or MQ wave will excite a QNM, but the latter will radiate as a combination of ED and MQ. Thus, the $R$-matrix is no longer diagonal, and energy can leak from one channel to another. At some critical values of $r_\perp/r_\parallel$, this results in the appearance of super dipoles. For instance, when the spheroid is oblate, the ED channel at $\omega_d$ almost doubles its allowed bound (Fig. 4b), while we observe the same effect at $\omega_u$ when the spheroid becomes prolate. Thus, depending on the sign of the deformation, we can enhance dipole scattering beyond the limit in one resonance or the other. This can also be confirmed in Figure S5 of the Supplementary Information, where the super-ED resonance is seen to exceed the single-channel limit, in contrast to the ED resonances of the perfect sphere.

We gain insight into the mechanism by evaluating the influence of each QNM to the ED scattering cross section (Fig. 4c, d). We consider three contributions: the resonant QNMs $|u,d\rangle$ and a non-resonant background composed of modes outside the spectral range of interest. With the expressions of the multipoles given in the Supplementary

Information S4, the ED cross section is

$$\sigma_{ED} = \left| \sum_m \boldsymbol{p}_m \right|^2 / I_0 \qquad (8)$$

$\boldsymbol{p}_m$ is the ED moment of the m-th QNM, and $I_0$ is the intensity of the incident plane wave. The reconstruction is in excellent agreement with the exact analytical results of Mie theory for the sphere (Fig. 4c), and full-wave numerical simulations for the ellipsoid (Fig. 4d). It is important to note that the 'direct' cross section of each QNM, (i.e., $|\boldsymbol{p}_m|^2/I_0$) by itself, is not bounded by any limit. However, there is a bound in the total ED cross section, as given by Eq. (8), in the case of the sphere.

In the upper panel of Fig. 4c, the scattering cross section at the resonance of $|u\rangle$ is clearly bounded to $3\lambda^2/2\pi$. However, once the sphere is deformed to an ellipsoid with $r_\perp/r_\parallel = 0.85$, $|d\rangle$ inherits an ED

moment due to coupling, and manifests as a sharp peak in the ED scattering cross section (Fig. 4d, upper panel), which gives rise to a super dipole.

The lower panels in Fig. 4c, d show the eigen-wavelengths $\tilde{\lambda}_m = 2\pi c/\tilde{\omega}_m$ of the two resonant QNMs in the complex plane, as well as their field distributions. For $r_\perp/r_\parallel = 1$, $\tilde{\lambda}_d$ is very close to the real axis. Once the spherical symmetry is broken, the additional radiative losses to the ED channel push $\tilde{\lambda}_d$ deeper into the complex plane, as originally predicted in the previous section. In addition, the hybridization between the two QNMs leads to a drastic reshaping of the internal field distributions of QNM $|d\rangle$ (compare the fields in the lower panels of Fig. 4c, d).

One important drawback of conventional superscattering is the fast degradation of the effect with intrinsic losses. Indeed, spectrally overlapping high order multipole resonances strongly maximizes scattering, but only when Ohmic losses are negligible. Unfortunately, even high-index dielectrics display small Ohmic losses in the visible, since their refractive index $n_p$ features a small imaginary part $\delta$. In general, we will write it as $n_p = n + i\delta$. As we show in the Supplementary Information S5, large Q-factors imply a rapid drop of the cross section maxima, yielding a slope for small $\delta$ of $d\sigma/d\delta \sim -4Q/\omega_0$, where $\omega_0$ is the resonance frequency. Since high-order multipole resonances are associated with large Q-factors, their maximum scattering cross section decreases rapidly with increasing $\delta$. Herein the reason why almost a decade passed since the original proposal until the experimental demonstration of superscattering[14].

As an example, consider the quasi-BIC resonance for the sphere case, displayed in Fig. 5a (indicated by the dashed line). In the lossless scenario, a strong scattering peak can be observed, reaching the maximum allowed for the MQ, i.e., $5\lambda^2/2\pi$. Thus, if one is only interested in the overall scattering cross section, there is no apparent need to 'transform it' into a super-dipole resonance, since the quasi-BIC already provides a significant scattering enhancement beyond the dipole limit. Moreover, by adding a shell of a different material, one could spectrally overlap the quasi-BIC with, e.g., the electric dipole, to yield a large enhancement. However, there is a caveat: increasing $\delta$ of the sphere by only 0.03 results in a drastic drop of the scattering cross section by more than 80%, even below that of a conventional dipole resonance. Therefore, in a practical scenario, high-order multipole resonances are not ideally suited to deliver the desired cross section.

Critically, super-multipole resonances, when formed through the FW mechanism, offer the ability to control the Q-factor. As discussed earlier, if the QNM is compatible with two scattering channels, the radiation losses increase, but contrarily to common belief, the total scattering cross section is not degraded, and can even increase at a super-multipole resonance. As a result, these novel states are more resilient to intrinsic losses, since the slope $d\sigma/d\delta$ is smaller than the original uncoupled resonances. Figure 5b illustrates this with the example of the super-ED resonance. The sphere of Fig. 5a is deformed into an ellipsoid, and the quasi-BIC evolves into a super-ED with lower Q-factor (dashed line in Fig. 5b). In order to make a fair comparison, the volume of the nanocavity is kept constant. In stark contrast with the quasi-BIC, the drop in the cross section is appreciably smaller, and for $\delta = 0.03$ it still remains above the single-channel limit.

A comparison between the maximal cross section attained by the quasi-BIC and the super-ED with increasing $\delta$ is displayed in Fig. 5c. For small $\delta$, we confirm that the quasi-BIC maximum has a much steeper slope as a function of $\delta$, while from the start, the super-ED keeps a much higher cross section. Thus, due to their inherent robustness to losses, super-multipole resonances are better candidates to enhance the scattering cross section at the nanoscale.

## Scattering from a dielectric nanorod

The previous example illustrates nicely the formation of a super dipole from a symmetry-breaking perturbation. However, spheroids are in general not fabrication-friendly at the nanoscale. Instead, we can also reach this regime in a similar fashion in a silicon nanorod under normally incident illumination (refer to inset of Fig. 6a), since the latter also has cylindrical symmetry.

To do so, we perturb the height of the resonator by an amount of $\Delta h$, starting from a height of $h_0 = 180$nm, for which two modes radiating as ED and MQ are spectrally close. As in the spheroid, we obtain a system of two coupled resonant QNMs of relatively high Q-factor. We also remark the presence of two additional QNMs of very low $Q$, associated with the scattering background. The role of the background modes is disregarded in most analysis since their spectral signature is barely appreciable. However, the correct eigenfrequencies and the scattering response cannot be accurately reconstructed without taking them into account.

In Fig. 6a the Q-factors of the resonant QNMs display two peaks as a function of $\Delta h/r$. The most pronounced one corresponds to the

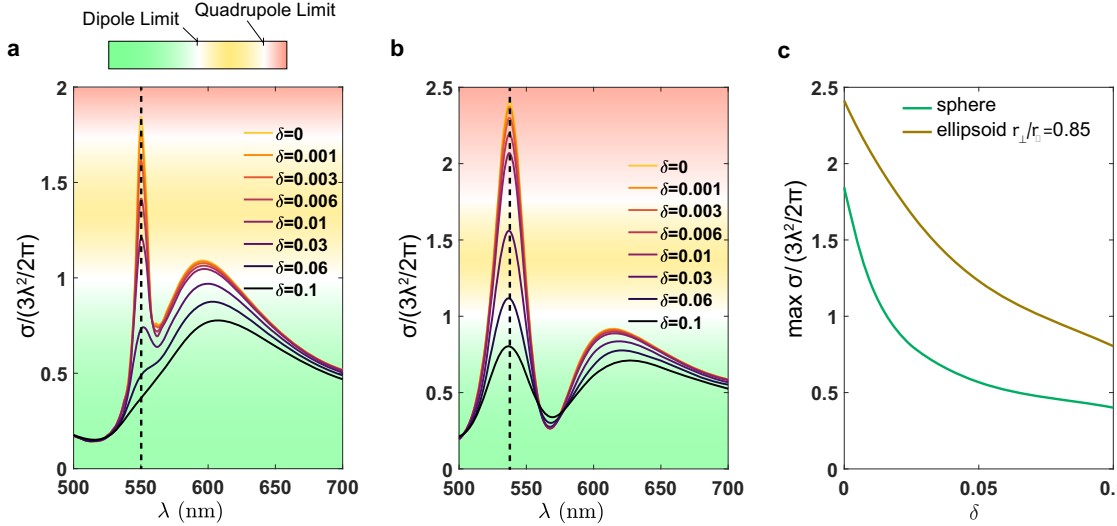

**Fig. 5 | Robustness of super-multipole resonances to Joule losses. a** Evolution of the total scattering cross section with increasing imaginary refractive index $\delta$ for a sphere and **b** an ellipsoid with $r_\perp/r_\parallel = 0.85$. The two scatterers have an equal volume and real index $n_p = 3.87$. **c** Comparison of superscattering maxima [dashed lines in **a** and **b**] with increasing losses for the quasi-BIC (sphere) and the super-dipole resonance (spheroid).

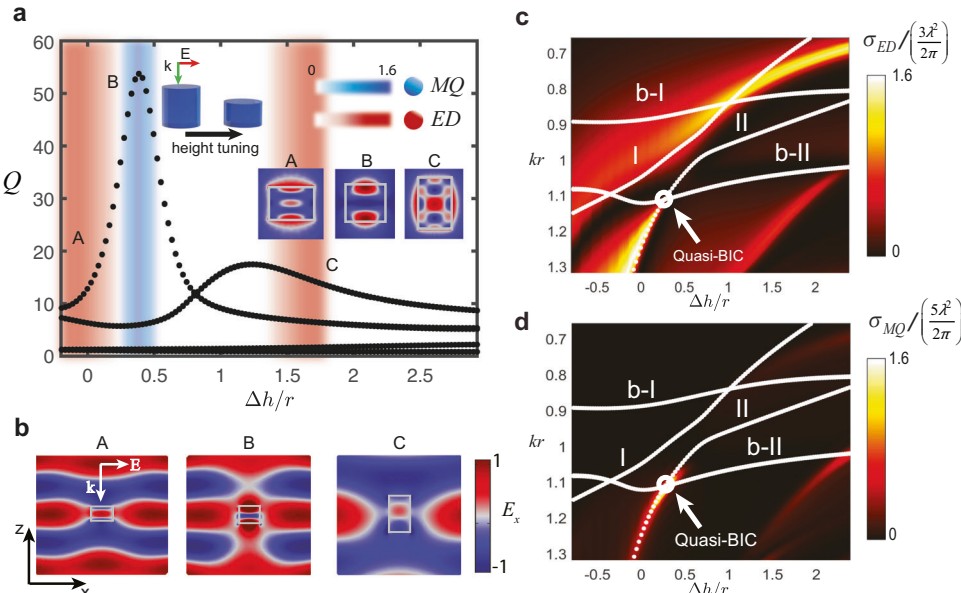

**Fig. 6 | Design of a super ED resonance in a Si nanorod. a** Quality factors and multipolar radiation of the QNMs as a function of a perturbation of the cylinder height $\Delta h$, normalized by the radius $r = 130$ nm. The color-bars indicate the predominant multipole. The artistic inset depicts the illumination scheme and the deformation undergone by the resonator. Rightmost panels A, B, C: electric field norms in the $x$–$z$ plane of the QNMs at points A, B, and C. **b** Shows the $E_x$ component of the total field at the superscattering points A-C indicated in **a**, under $x$-polarized plane wave illumination (in arbitrary units). **c, d** 2D maps of the ED and MQ scattering cross-sections as a function of $kr$ and $\Delta h/r$. White dotted lines indicate the paths followed by the eigenfrequencies, obtained with the perturbation theory derived in section S1 of the Supplementary Information. I,II correspond to the resonant QNMs, and b-I, b-II are the background ones. In agreement with theory, departing from the quasi-BIC condition results in hybridization and the possibility to enhance scattering of a single mode beyond the limit.

hybridization of the resonant QNMs, while the second is due to the hybridization of a resonant QNM with the low-$Q$ modal background. Superscattering in the ED channel (a super dipole) arises in the red-shaded regions. Interestingly, the super dipole appears for both resonant QNMs at relatively low $Q$-factors (points A and C). Near the quasi-BIC (point B), the dipole strength becomes quenched and the radiation leaks solely through the MQ channel, (blue-shaded region) with high-$Q$. This peculiarity can also be clearly seen in the 2D maps of the ED and MQ cross-sections shown in Fig. 6c, d. In points A and C, the most appreciable signature of ED radiation is the appearance of a central electric field hotspot inside the nanorod, together with side lobes. As shown in the field insets of Fig. 6a, at the quasi-BIC, the hotspot disappears. This behavior is exactly analogous to what took place in the nanoellipsoid from Fig. 4. In all cases, the incident plane wave is significantly distorted by the scattered field (Fig. 6b).

In the super dipole regime, the ED is shown once again to almost double its established bound (Fig. 6c). Similarly, we observe a peak in the $Q$-factor (Fig. 6a) and an enhanced MQ scattering cross section (Fig. 6d) at the quasi-BIC. Thus, we have demonstrated the feasibility to obtain super dipoles in an experimentally accessible platform. This result reveals a new versatile strategy that can be used to engineer the $Q$-factor, scattering efficiency and radiation pattern of an isolated, subwavelength object in practical applications.

## Experimental demonstration
We perform a proof-of-concept experiment by measuring the extinction cross-section and scattering patterns of disk-shaped resonators in the microwave range. We reproduce the geometrical parameters of the rod in Fig. 6 using a set of ceramic resonators with fixed 4.0 mm radii, and permittivity $\varepsilon = 22$ with loss tangent 0.001. As shown in the inset of Fig. 7b, three samples are assembled from several disks to obtain the desired aspect ratios for the resonators. The measurement results of both the total extinction cross-section and electric near-field patterns are collected in Fig. 7 (for more details in the experiment, refer to section S8 of the Supplementary Information). The spectra are

normalized by the ED single-channel limit ($\sigma_{Max} = 3\lambda^2/2\pi$). The experimental measurements are in a reasonable agreement with the numerical simulations, albeit the resonances appear suppressed due to material losses in the ceramic. Even with the latter, the ED cross section at the super ED is still significantly higher than the limit, as we show in Figure S3 of the Supplementary Information.

Since the resonances redshift with increasing size, the observations were performed in a broad frequency range. In the highlighted frequencies of Fig. 7a, c, we observe wide resonances with large extinction values, characteristic of the proposed super dipole modes. Indeed, the plane wave is seen to be strongly distorted in the near field (lower panels of Fig. 7). Furthermore, numerical calculations confirm that the ED exceeds its limit, even when considering losses, (refer to Fig. S3 of the Supplementary Information). The quasi-BIC appears at the expected value of $\Delta h/r = 0.48$, manifesting itself as a sharp peak in the spectra (Fig. 7b). The results provide experimental evidence of the control of both the $Q$-factor and scattered power between two resonances to achieve the superscattering regime with just a single mode.

## Boosting a super-multipole even further
In what follows, we discuss the possibility to enhance the cross-section of a single multipole beyond what has been achieved so far. For that purpose, we study a subwavelength rectangular prism with refractive index $n = 3.3$ in the near-IR part of the spectra (inset of Fig. 8a). The prism has unequal sides in all $x, y,$ and $z$ dimensions. Starting from an initial prism, we vary the y and z sides by the same amount $\delta$, keeping the $x$ side constant. Tuning $\delta$ towards negative or positive values results in the appearance of the quasi-BIC or the super MD resonance. In both cases, a drastic reshaping of the near fields takes place, particularly pronounced in the case of the lower (high-$Q$) branch (Fig. 8d).

As in the previous cases, we identify two coupled QNMs $|u\rangle$ and $|d\rangle$ (Fig. 8a). Now, however, the $Q$-factor at the quasi-BIC is two times larger. This is because the new quasi-BIC is associated not with a quadrupole, but with a pure magnetic octupole response, similar to the one investigated in[5]. On the other hand, QNM $|d\rangle$ has a

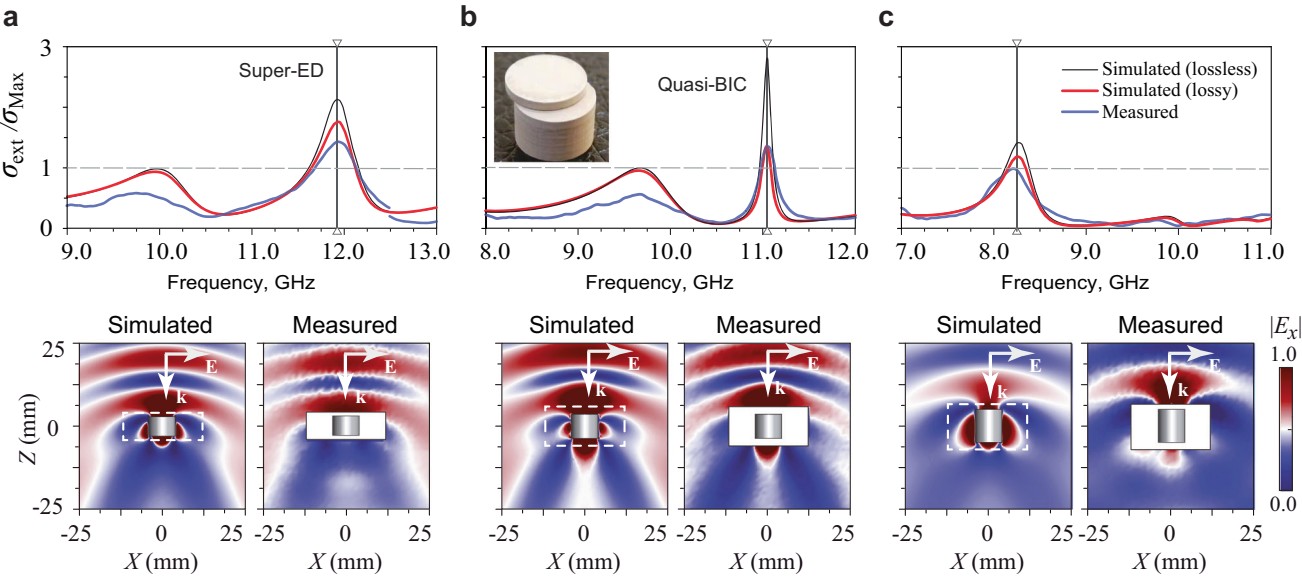

**Fig. 7 | Experimental confirmation of super-ED resonances.** Simulated and measured total extinction cross-sections and scattered electric near-field patterns. All the cross-sections are normalized by the single-channel limit for the ED, $\sigma_{Max} = 3\lambda^2/2\pi$. Insets show an example of the experimental resonator and the electric field norms in the $x$–$z$ plane of the resonances indicated by vertical lines in the top plots. The white regions in the near-field patterns correspond to the physically inaccessible zones for the measurements. The aspect ratios of the disks are: **a** $\Delta h/r = 0.25$, **b** $\Delta h/r = 0.475$, and **c** $\Delta h/r = 1.25$.

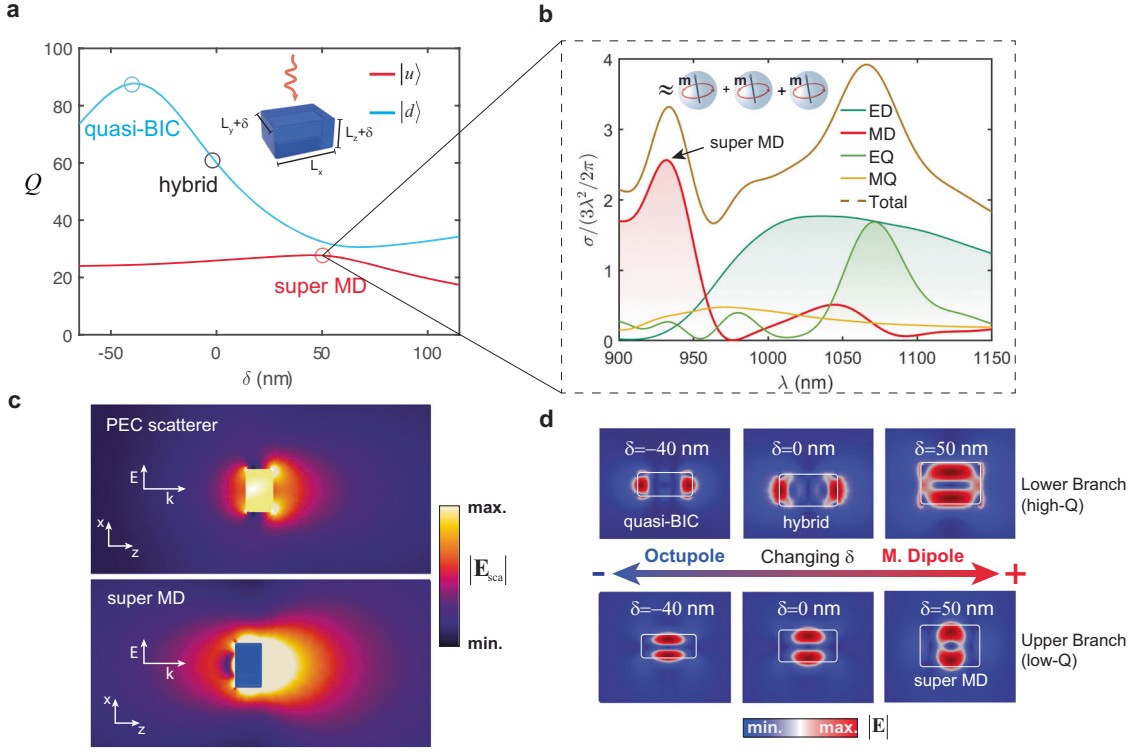

**Fig. 8 | Super-MD resonance produced by dipole-octupole hybridization. a** Inset: scheme depicting the geometry of the cavity and the incident illumination. The initial cavity is a rectangular prism with unequal sides $L_x = 500$nm, $L_y = 435$nm, $L_z = 260$nm, and refractive index $n_p = 3.3$. $L_x$ is kept constant, while $L_y$ and $L_z$ are progressively tuned by an amount $\delta$. $Q$-factors of the two involved QNMs as a function of $\delta$. The quasi-BIC appears in the lower branch (blue circle), characterized by an overall high $Q$-factor, for $\delta = -40$ nm, while the super MD appears at the upper branch (with overall low $Q$-factor) for a detuning of $\delta = 50$ nm from the initial geometry (red circle). It coincides with a slight enhancement of the $Q$-factor of the upper branch and a decrease in the lower one. **b** Multipole decomposition of the scattering cross section for the cavity with $\delta = 50$ nm, normalized by the dipole

limit. We observe a remarkable enhancement of the MD cross-section of >2.6 times the conventional limit. The latter is almost equivalent to the cross section of three magnetic resonances of isolated spheres at the same wavelength (see inset). Incidentally, we also observe the accidental overlap of the high-$Q$ branch, radiating as a combination of electric quadrupole (EQ) and MD, with a super ED resonance similar to the nanorod in the previous section. **c** Comparison between the scattered fields produced by a perfect electric conductor (PEC) cavity (upper panel) and the designed superscatterer (lower panel), both with the same dimensions. The fields are recovered at the wavelength of 935 nm, corresponding to the large MD peak in **b**. **d** Near-field distribution of the involved QNMs for selected $\delta$ (refer to discussion in text).

predominant magnetic dipole (MD) response. When changing $\delta$, we observe the emergence of a strong MD peak, with a cross section equivalent to almost three magnetic spheres (Fig. 8b). Interestingly, the super MD resonance appears in the $|d\rangle$ branch, which does not feature the quasi-BIC. This is not surprising since the branches are coupled and as discussed earlier, super-multipole resonances are not generally restricted to appear in either one of the two.

The reason for the stronger enhancement can be understood from noticing that, according to Eq. (1), the magnetic octupole with $l=3$ is bounded to $\sigma_{Max} = 7\lambda^2/2\pi$, while the magnetic quadrupole bound is $5\lambda^2/2\pi$. Thus, the higher the order of the multipole that couples to the dipole, the larger the enhancement of the dipolar cross section that can be achieved. In addition, numerical and experimental evidence has shown that quasi-BICs with high order multipole response have an increasingly larger $Q$-factor[37]. It is possible to utilize this fact as a handwaving design rule for super-multipole resonances: the larger the $Q$-factor at the quasi-BIC, the larger the potential enhancement of the low order multipole.

To visualize the strong scattering response of the super MD resonance, we compare it with a rectangular prism of the same size but composed of perfect electric conductor (PEC). Figure 8c shows the amplitude of the scattered electric field in both cases. It is worth noting that a significant enhancement can be appreciated for the dielectric prism, both in the forward and in the backward directions. This constitutes an important difference with respect to conventional superscattering. In the vast majority of designs only forward scattering can be maximized[14,18,23,26]. Thus, super-multipole resonances offer an attractive strategy to enhance backscattering without the need to sacrifice the overall scattering efficiency, as is the case for the anti-Kerker effect[38].

In principle, the strategy above allows to enhance scattering by a single multipole to arbitrarily large values. Another approach is to combine super-multipoles with Fan et al.'s original method[16]. Namely, we can spectrally overlap the super-multipole resonance with other conventional resonances. As a proof-of-concept, we designed a dielectric nanocylinder with $n_p = 3.3$ where a super-ED accidentally crosses with the MD resonance as a function of height. The combination of both resonances leads to very large cross-sections, in the order of 5 times the single-channel limit [Fig. 9b]. This result is more than three times what can be achieved when overlapping the conventional ED and MD resonances [Figure S4c in the Supplementary Information]. Moreover, we notice that 70% of the enhancement is entirely due to the super-ED.

Figure 9c shows simulations of the strong field distortion produced by such superscatterer. The latter leaves a large 'shadow' where field intensity is significantly lowered. In addition, Fig. 9d shows a comparison between the far fields of our superscatterer and a PEC cavity of the same dimensions. The superscatterer clearly exhibits superior performance, displaying enhanced forward and backward scattering.

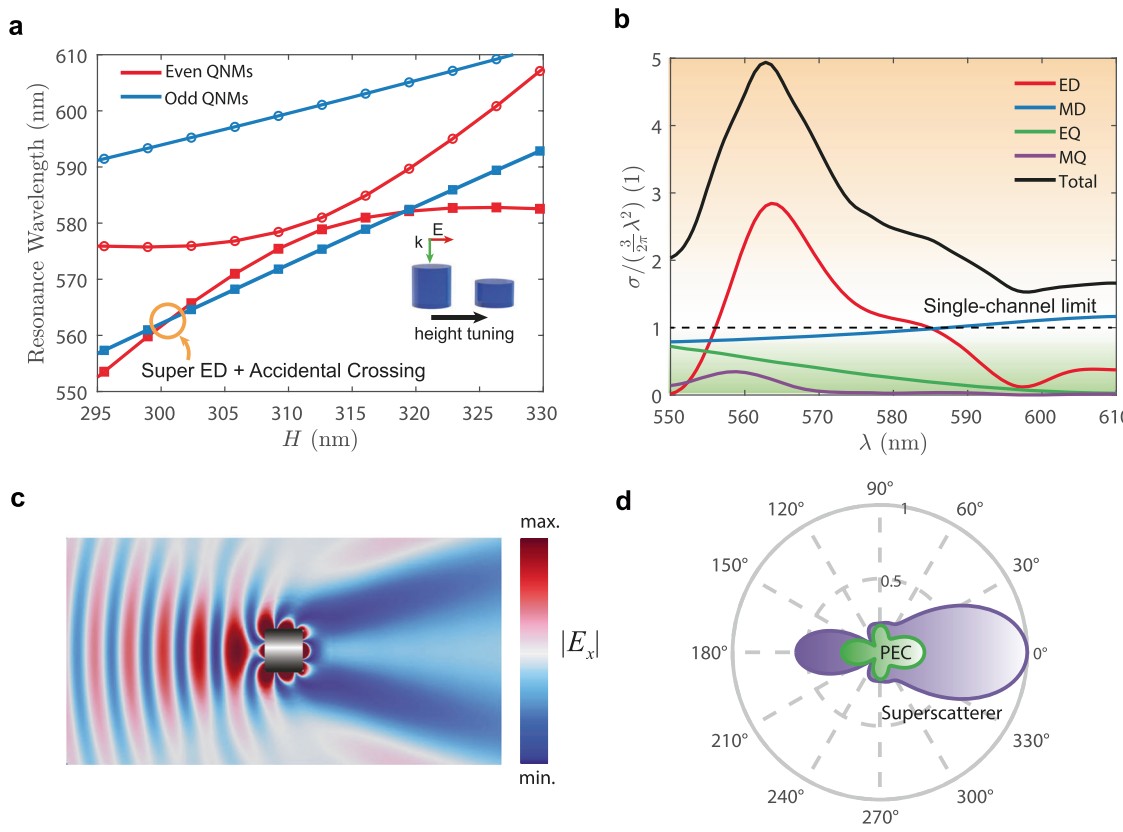

**Fig. 9 | Combining the old and the new mechanisms of superscattering.** Combining the old and the new mechanisms to achieve even more superscattering from passive, subwavelength nanostructures. **a** Calculated resonance wavelengths of the even and odd QNMs for a dielectric nanocylinder with $n_p = 3.3$ and radius 160 nm. The even QNMs are spectrally close and they couple, resulting in the formation of a quasi-BIC and a super-ED as a function of height ($h$). Now, however, we designed the nanoparticle such that the super-ED spectrally overlaps with an odd QNM (accidental crossing), following the original strategy by Fan et al.[16]. Squares and dots denote the dispersions of different modes. **b** Scattering cross section for $h = 300$ nm. The super-ED boosts the ED cross-section by three times the single-channel limit. In combination with the MD and higher-order multipolar contributions, the total cross-section reaches five times the limit. **c** x-component of the electric field at $\lambda = 565$ nm. The field can be seen to be strongly distorted by the scatterer. **d** Comparison between the far fields produced by a perfect electric conductor (PEC) cavity (green pattern) and the designed superscatterer (purple pattern), both with the same dimensions. The fields have been normalized to the maximum of the superscatterer.

## Shielding nanoparticles from scattering forces

Finally, we provide a glimpse on the possibilities that can be unlocked in optics with the realization of the superscatterers introduced in this study. As an exemplary application, we propose a strategy to 'protect' an ensemble of nanoparticles (or one particle, but geometrically significantly larger than the superscatterer) from radiation. In particular, as a figure of merit we will consider parasitic scattering forces induced by an incident beam. Reducing the influence of scattering forces is essential for efficient optical traps for experiments in atom cooling or modern biology[39,40].

Figure 10a illustrates the main idea. Due to the strong scattering, the Poynting vector lines (purple) near the superscatterer are strongly distorted, leading to a large 'shadow' area behind it. Several scatterers can be 'hidden' in the shadow, which significantly reduces the scattering force experienced by them (red).

In particular, we consider the scatterers can be modeled as point electric dipoles with an effective polarizability $\alpha$. In the case under consideration, the dominant scattering force experienced by dipolar particles is given by[41] $F_z \propto Im(\alpha)S_z$, where $S_z$ is the z-component of the Poynting vector. The black arrows in Fig. 10b show the distribution of $F_z$ as a function of position near the superscatterer presented in Fig. 9. Inside the shadow region, the latter can be seen to be strongly suppressed. Figure 10c shows the calculated ratio of optical force with and without the superscatterer at a fixed height. Remarkably, scattering forces can be decreased in a region much larger than the diameter of the superscatterer. Hence, several scatterers can be simultaneously hidden in the shadow.

## Discussion

We have demonstrated how strong coupling of two resonances can be harnessed to achieve novel superscattering regimes with sub-wavelength, nonspherical resonators. We have observed super-scattering originating from an electric super dipole moment, being almost two times stronger than the currently established limit. In resonators without spherical symmetry, this effect arises when breaking the quasi-BIC condition by tuning some parameter. Then, power exchange between the scattering channels allows to engineer both Q-factors and multipolar contents of the resonances, while maintaining a high scattering cross-section. The new super-multipole resonances are more robust to Ohmic losses than their conventional counterparts. Furthermore, we have shown how the enhancement can be boosted even further when quasi-BICs associated with high order multipoles are involved. This enables the formation of super magnetic dipole moments with a cross-section equivalent to almost three magnetic spheres. Besides their fundamental interest, such exotic scattering can be employed in biosensing[42,43] or energy harvesting[44–46] devices. In the near future, strongly scattering dielectric nanoantennas, operating in a selective polarization regime, can replace their plasmonic counterparts as ultra-compact demultiplexers for on-chip circuitry[47]. Furthermore, the ability to selectively enhance the scattering pattern of a given multipole can unlock new degrees of freedom for optical manipulation[48,49]. In this direction, by taking advantage of the designed superscatterer, we have proposed a new strategy to shield an ensemble of particles from radiation, namely—parasitic scattering forces. Beyond optics, we expect super-multipoles to also

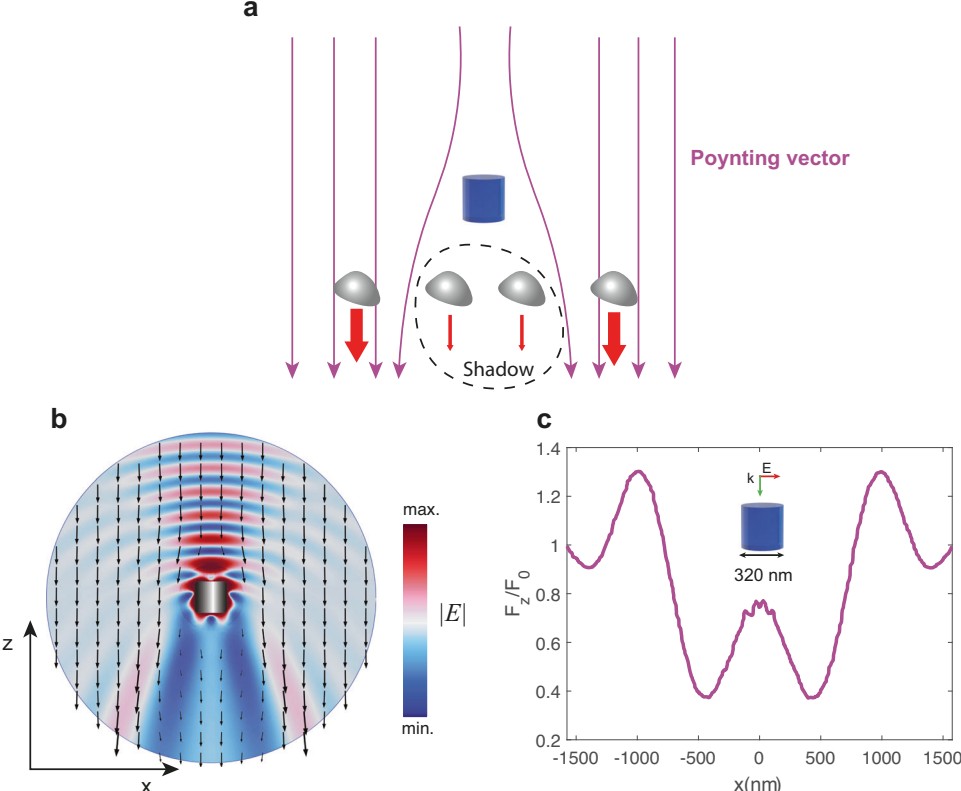

**Fig. 10 | Protecting other scatterers from radiation. a** Conceptual scheme. The superscatterer interacts with photons in a much larger area than itself. As a result, the Poynting vector field lines (purple arrows) are deflected, and the superscatterer leaves a large 'shadow', much larger than its diameter[16]. The scatterers placed within that shadow (gray shapes) are 'protected' from the radiation pressure (red arrows) induced by the incident beam. **b** Electric field norm in the vicinity of the superscatterer (same parameters as in Fig. 9), and calculated radiation pressure experienced by dipolar particles (black arrows). The scattering force (and, consequently, visibility) is significantly decreased within the shadow. **c** Ratio between the scattering force with and without the superscatterer, experienced by dipolar particles positioned at a distance $z = -1200$ nm from the superscatterer. The origin of coordinates is at the position of the superscatterer. Inset: artistic view of the superscatterer, scaled to match the grid dimensions of the $x$ axis.

arise in acoustics and other areas of wave physics, paving a way to a venue of applications in strong forward or backward scattering, cloaking, energy harvesting, etc.

## Methods

### Temporal coupled-mode theory

General predictions on the interaction of the scattering channels with the QNMs can be made through a widely applicable phenomenological theory known as the TCMT. The formalism was originally used to introduce the concept of superscattering[16]. Here, we briefly introduce the theory, (whose details can be found in a number of seminal works, e.g., [50,51]. The coupled mode equations can be written as

$$\frac{d}{dt}\begin{pmatrix} a \\ b \end{pmatrix} = -i\mathcal{H}_0(\zeta)\begin{pmatrix} a \\ b \end{pmatrix} + i\sqrt{2}D^T \boldsymbol{s}^+ \qquad (9)$$

and

$$\boldsymbol{s}^- = \boldsymbol{s}^+ + i\sqrt{2}D\begin{pmatrix} a \\ b \end{pmatrix} \qquad (10)$$

For the derivation of the effective Hamiltonian $\mathcal{H}_0$ as in Eq. (2), we refer the reader to the Supplementary Information S4. In addition, arguments based on time reversal symmetry and energy conservation constraints[51] lead to the relation

$$D^T D = \Gamma \qquad (11)$$

where $\Gamma = -Im\{\mathcal{H}_0\}$ is a diagonal matrix containing the radiative losses $\gamma_{a,b}$ in its diagonal. Equation (11) implies that $\gamma_{a,b} = d_{1,2}^2$, $d_i$ being the i-th element in the diagonal of $D$. It accounts for the coupling of the eigenmodes to the i-th multipole channel. The $R$-matrix, (Eq. (4) in the main text), can be derived by assuming time-harmonic dependence [$d/dt \rightarrow -i\omega$ in Eq. (9)], and substituting Eq. (9) into Eq. (10) to eliminate $\begin{pmatrix} a & b \end{pmatrix}^T$. Further details on the connection between TCMT and rigorous QNM perturbation theory can be found in the Supplementary Information S2.

### Numerical simulations

Scattering and eigenmode simulations have been performed with the commercial finite element solver COMSOL Multiphysics ©.

### Experimental methods

A Taizhou Wangling TP-series microwave ceramic composite is used as a dielectric material for the fabrication of the cylindrical resonators. To measure the total extinction cross section, the samples are placed in an anechoic chamber and illuminated by normally incident, linearly polarized waves radiated and received by a pair of HengDa Microwave HD-10180DRA10 horn antennas. A LINBOU near-field imaging system is used for the near-field mapping. More details on the fabricated samples and the experimental setup can be found in the Supplementary Information S8 and Figure S2.

## Data availability

All data needed to evaluate the conclusions in this study is presented in the manuscript and in the Supplementary Information.

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

## Acknowledgements

The research was supported by Priority 2030 Federal Academic Leadership Program. The authors gratefully acknowledge the financial support from the Ministry of Science and Higher Education of the Russian Federation (Agreement No. 075-15-2022-1150). The calculations of quasi-BICs and super-multipoles are partially supported by the Russian Science Foundation grant o. 23-72-00037. A.S.K. acknowledges financial support from the National Key R&D Program of China (project No. 2018YFE0119900). A.C.V. gratefully acknowledges funding from the Russian Science Foundation project No. 22-42-04420, for the calculation of QNMs of the nanorod. Y.K. acknowledges support from the Australian Research Council (grant DP210101292). V.B. acknowledges the support of the Latvian Council of Science, project: NEO-NATE, No. lzp-2022/1-0553.

## Author contributions

A.C.V. and H.K.S. conceived the idea. A.C.V. developed the theory and wrote the first draft of the manuscript. A.C.V. performed the initial numerical simulations. H.K.S conducted numerical optimizations of the nanoscatterers. A.S.K. performed the experiment. A.S. and Y.K. extended the original idea, supervised the project, proofread the manuscript, and actively participated in the discussions. T.W. proofread the final version of the manuscript and provided valuable suggestions. A.A.P., V.B., and D.R. contributed to the writing of the manuscript and the discussions.

## Competing interests

The authors declare no competing interests.
