## [Peer review file · Nature Communications]

REVIEWER COMMENTS

Reviewer #1 (Remarks to the Author):

The authors propose a new mechanism for superscattering with subwavelength nanoresonators. With both the theoretical and the experimental studies, they state that the scattering cross section can overcome the single channel limit when avoiding crossing resonances associated with quasi-BIC states. Despite some potential advance, there are still some concerns that need to be addressed prior to any decision.

1. As Eq.(2), the single channel limit in the three-dimensional case is associated with l . For the dipole and the quadrupole channel, they are $3\lambda^2/2\pi$ and $5\lambda^2/2\pi$, respectively. Since the quasi-BIC corresponds to the MQ mode, the limit of σ_0 in Fig.2-5 should be $5\lambda^2/2\pi$, instead of $3\lambda^2/2\pi$. It needs more clear explanation why labeling such mode with “dipole”. Otherwise, all these results would be still below the single-channel limit.

2. Although the superscattering is emerging when tuning the geometry parameters from the quasi-BIC case in the present study, if I understand correctly, generally the realization of superscattering does not require the quasi-BIC mode. That is, the quasi-BIC mode cannot directly and sufficiently lead to the superscattering by tuning the geometry parameters. In this sense, what association of the quasi-BIC with the superscattering?

3. (Line 5 of the second paragraph in Page 5) $|a, b\rangle$ should be written as $|a\rangle$ and $|b\rangle$. Also below Eq.(2) $|u, d\rangle$ should be written as $|u\rangle$ and $|d\rangle$.

Reviewer #2 (Remarks to the Author):

In this work, the authors proposed another superscattering method by strong coupling of the resonant modes. Compared with widely used superscattering effect, the main difference lies in the breaking of symmetric structure. This way, the scattering strength can be enhanced due to constructive interferences. Also, the authors provided a general non-Hermitian model to describe interfering resonances of quasi-normal modes, and experimentally confirmed the theoretical findings. Some important problems must be addressed before I could give the paper a further consideration.

1) If ignoring the symmetric requirement, by further optimizing the structure (not a cylinder with finite length), more interesting scattering phenomena and more scattering enhancement will be found. Probably this is a natural result by breaking the spherical symmetry to induce other interesting phenomena. We suggest the authors investigate other peculiar and asymmetrical structures, assisted with optimization algorithm, to assess the scattering performance.

2) We have seen a similar nanodisk superscattering structure before (Nanoscale 6, 9093–9102 (2014) & Phys. Rev. Lett. 112, 113903 (2014)). What are the difference and advantage compared with these literatures.

3) From the result in Fig. 5, the scattering cross section only slightly exceeds the single-channel limit (about 1.5 times). I suggest the authors give us more convincing results in the presence of loss. Also, I suggest the authors compare the scattering performance with other object, such as a PEC disk with the same geometry.

4) The authors stress that “our mechanism is in stark contrast with the conventional way to achieve superscattering”. I find the underlying mechanism is the cross of two modes to create a super dipole mode. In this regard, the working mechanism is similar to some extent.

5) Please state the practical significance and application of conventional superscattering and symmetry breaking superscattering in detail. It would be better to show several specific examples, beyond the theoretical study.

6) In my understanding, superscattering is a device to help other objects enhance their scattering, just like invisibility cloak that wraps up other objects. I am wondering superscattering cannot realize this goal because it only enhances itself scattering. How to understand this point?

Reply to Reviewers' Reports and a Summary of the Changes made in the Revised Manuscript

Reply to Reviewer #1

Reviewer #1

1. As Eq.(2), the single channel limit in the three-dimensional case is associated with l . For the dipole and the quadrupole channel, they are $3\lambda^2/2\pi$ and $5\lambda^2/2\pi$, respectively. Since the quasi-BIC corresponds to the MQ mode, the limit of σ_0 in Fig.2-5 should be $5\lambda^2/2\pi$, instead of $3\lambda^2/2\pi$. It needs more clear explanation why labeling such mode with "dipole". Otherwise, all these results would be still below the single-channel limit.

Our reply

First, we notice that the limitation applies to the scattering channels (provided by multipoles), but not the modes. Thus, it is the scattering cross section of each multipole (not mode), what is generally considered that cannot exceed the limit.

The quasi-BIC radiates only to the MQ channel and is indeed constrained by $5l^2/2\pi$. Similarly, a mode radiating as an ED is constrained to $3\lambda^2/2\pi$.

However, we study the condition when the quasi-BIC is hybridized with a mode radiating to the dipole channel, so that energy that would originally radiate to the MQ can be redirected to the dipole channel. Through this process, the quasi-BIC gains dipolar character, reduces its quality factor, and the ED cross section associated with the resonance of the mode can be maximized beyond $3\lambda^2/2\pi$.

We noticed that Figure 4d (now Figure 6d), presented the MQ cross section normalized by the dipole limit, which might induce confusion. In the new version we have normalized it by its proper limit, namely $5\lambda^2/2\pi$, as suggested by the Reviewer. Along the same lines, we have avoided the notation of ' σ_0 ' to denote the limit in all parts of the text where it might result in misunderstandings for the readers.

Figure 6. Design of a super dipole mode in a Si nanorod. (a) Quality factors and multipolar radiation of the QNMs as a function of a perturbation of the cylinder height Δh , normalized by the radius $r=130$ nm. The color-bars indicate the predominant multipole. The artistic inset depicts the illumination scheme and the deformation undergone by the resonator. Rightmost panels A,B,C: Electric field norms in the x - z plane of the QNMs at points A, B, and C. (b) Shows the E_x component of the total field at the superscattering points A-C indicated in (a), under x -polarized plane wave illumination (in arbitrary units). (c)-(d) 2D maps of the ED and MQ scattering cross sections as a function of kr and $\Delta h / r$. White dotted lines indicate the paths followed by the eigenfrequencies, obtained with the perturbation theory. I,II correspond to the resonant QNMs, and b-I, b-II are the background ones. In agreement with theory, departing from the quasi-BIC condition results in hybridization and the possibility to enhance scattering of a single mode beyond the limit.

Finally, we have added Figure 1b (shown below), displaying the scattering cross section and multipole decomposition at the super dipole resonance for the Si cylinder studied in the main text. From this result, the Reviewer can appreciate that the scattering cross section of the ED indeed exceeds the limit.

Figure 1. (a) Concept of BIC-inspired superscattering in an isolated resonator. Strong coupling of two modes reshapes both their near fields and scattering patterns as a function of a tuning parameter. The interfering resonances lead to a quasi-BIC state (destructive interference) and induce power redistribution between multipolar scattering channels leading to super-dipole radiation (constructive interference). (b) Super dipole resonance arising in the scattering cross section of a dielectric cylinder with refractive index $n \approx 3.8$, radius 130 nm and height 180 nm. The scattering cross section of the electric dipole channel is seen to significantly exceed the single-channel limit. This is in contrast with conventional superscattering, where several multipole resonances need to be overlapped.

Reviewer #1

2. Although the superscattering is emerging when tuning the geometry parameters from the quasi-BIC case in the present study, if I understand correctly, generally the realization of superscattering does not require the quasi-BIC mode. That is, the quasi-BIC mode cannot directly and sufficiently lead to the superscattering by tuning the geometry parameters. In this sense, what association of the quasi-BIC with the superscattering?

Our reply

We thank Reviewer #1 for his/her thorough reading of the manuscript. In scatterers without spherical symmetry, both processes exist (superscattering and quasi-BIC). They are counterparts of the same mechanism of interfering resonances. A quasi-BIC occurs because of destructive interference of the modes radiating to the same channel, and this new superscattering effect occurs because the two modes interfere constructively in the same channel (in the absence of spherical symmetry). This interference results in new hybrid modes with potentially enhanced dipole scattering. Thus, to the best of our knowledge, the appearance of quasi-BICs is indeed a *sufficient* condition for the existence of super multipoles. However, it is not a *necessary* condition.

Figure 8. Super MD resonance produced by dipole-octupole hybridization. (a) Inset: scheme depicting the geometry of the cavity and the incident illumination. The initial cavity is a rectangular prism with unequal sides $L_x = 500$ nm, $L_y = 435$ nm, $L_z = 260$ nm, and refractive index $n=3.3$. L_x is kept constant, while L_y and L_z are progressively tuned by an amount δ . Q-factors of the two involved QNMs as a function of δ . The quasi-BIC appears in the lower branch (blue circle), characterized by an overall high Q-factor, for $\delta = -40$ nm, while the super MD appears at the upper branch (with overall low Q-factor) for a detuning of $\delta = 50$ nm from the initial geometry (red circle). It coincides with a slight enhancement of the Q-factor of the upper branch and a decrease in the lower one. (b) Multipole decomposition of the scattering cross section for the cavity with $\delta = 50$ nm, normalized by the dipole limit. We observe a remarkable enhancement of the MD cross section of more than 2.6 times the conventional limit. The latter is almost equivalent to the cross section of three magnetic resonances of isolated spheres at the same wavelength (see inset). Incidentally, we also observe the accidental overlap of the high-Q branch, radiating as a combination of electric quadrupole (EQ) and MD, with a super ED resonance like the nanorod in the previous section. (c) Comparison between the scattered fields produced by a perfect electric conductor (PEC) cavity (upper panel) and the designed superscatterer (lower panel), both with the same dimensions. The fields are recovered at the wavelength of 935 nm, corresponding to the large MD peak in (b). (d) Near field distribution of the involved QNMs for selected δ (refer to discussion in text).

While we do not claim that the Friedrich-Wintgen mechanism is the only strategy to realize a super dipole resonance, in practice, it is a very efficient way to do so (and the only one that we have found so far). Namely, as we now explain in the **new section** entitled ‘super-

multipole resonances, engineering a super dipole resonance requires a way to control the radiation rate of a QNM to two multipoles. We found that super multipole resonances occur only for a critical radiation rate.

Due to the Friedrich-Wintgen mechanism, the radiation rates of the QNMs change smoothly as a function of a single system parameter. What's more, at the quasi-BIC, the radiation rate to the lowest order multipole is, by definition, zero (c.f. Ref. *Advanced Photonics* 1.1 (2019): 016001). By varying the system parameter (e.g., the cylinder height) starting from the quasi-BIC, one can easily 'select' the critical radiation rate to the lowest order multipole that will lead to a super multipole resonance.

To demonstrate the generality of our findings, we have designed another Si nanocavity (a rectangular prism) featuring a super magnetic dipole (super MD) resonance arising from the Friedrich Wintgen mechanism (Figure 8, attached above). We note that the super MD now forms from the other QNM (Figure 8a). This is not surprising, since the radiation rates of both QNMs change as a function of the cavity dimensions.

Reviewer #1

3. (Line 5 of the second paragraph in Page 5) $|a, b\rangle$ should be written as $|a\rangle$ and $|b\rangle$. Also below Eq.(2) $|u, d\rangle$ should be written as $|u\rangle$ and $|d\rangle$.

Our reply

Thank you very much for pointing out the misprints, they are now corrected.

Reply to Reviewer #2

Reviewer #2

1) If ignoring the symmetric requirement, by further optimizing the structure (not a cylinder with finite length), more interesting scattering phenomena and more scattering enhancement will be found. Probably this is a natural result by breaking the spherical symmetry to induce other interesting phenomena. We suggest the authors investigate other peculiar and asymmetrical structures, assisted with optimization algorithm, to assess the scattering performance.

Our reply

Following the suggestion of Reviewer #2, we investigated another geometry with lower in-plane rotational symmetry, namely a rectangular prism. Here, we have found a quasi-BIC associated with an octupole response (Figure 8a). Remarkably, the coupling between this quasi-BIC and a magnetic dipole-like mode results in the formation of a super magnetic dipole resonance (super MD), featuring a scattering cross section exceeding by almost three times the limit of a single 'magnetic' sphere (Figure 8b), thus almost doubling the previous result. The new calculations prove the possibility to enhance at will the cross section of a specific multipole (either electric or **magnetic**), which can have important implications to optimize light-matter interactions, for instance to efficiently couple with magnetic emitters, or control the directionality of light in a waveguide.

Figure 8. Super MD resonance produced by dipole-octupole hybridization. (a) Inset: scheme depicting the geometry of the cavity and the incident illumination. The initial cavity is a rectangular prism with unequal sides $L_x = 500$ nm, $L_y = 435$ nm, $L_z = 260$ nm, and refractive index $n=3.3$. L_x is kept constant, while L_y and L_z are progressively tuned by an amount δ . Q-factors of the two involved QNMs as a function of δ . The quasi-BIC appears in the lower branch (blue circle), characterized by an overall high Q-factor, for $\delta = -40$ nm, while the super MD appears at the upper branch (with overall low Q-factor) for a detuning of $\delta = 50$ nm from the initial geometry (red circle). It coincides with a slight enhancement of the Q-factor of the upper branch and a decrease in the lower one. (b) Multipole decomposition of the scattering cross section for the cavity with $\delta = 50$ nm, normalized by the dipole limit. We observe a remarkable enhancement of the MD cross section of more than 2.6 times the conventional limit. The latter is almost equivalent to the cross section of three magnetic resonances of isolated spheres at the same wavelength (see inset). Incidentally, we also observe the accidental overlap of the high-Q branch, radiating as a combination of electric quadrupole (EQ) and MD, with a super ED resonance like the nanorod in the previous section. (c) Comparison between the scattered fields produced by a perfect electric conductor (PEC) cavity (upper panel) and the designed superscatterer (lower panel), both with the same dimensions. The fields are recovered at the wavelength of 935 nm, corresponding to the large MD peak in (b). (d) Near field distribution of the involved QNMs for selected δ (refer to discussion in text).

Reviewer #2

2) We have seen a similar nanodisk superscattering structure before (*Nanoscale* 6, 9093–9102 (2014) & *Phys. Rev. Lett.* 112, 113903 (2014)). What are the difference and advantage compared with these literatures.

Our reply

We thank Reviewer #2 for mentioning these two key works. For the following discussion we will label them as *Ref.1* and *Ref.2*, respectively.

- (1) In *Ref.1*, the authors claimed to realize superscattering by manipulating the geometrical degrees of freedom of a cylinder but did not demonstrate the possibility to *break the single channel limit* of the dipole, nor realized the potential of the *Friedrich-Wintgen mechanism* to control the enhancement, which are the two most important findings in our work. Moreover, all scattering cross sections in *Ref.1* are normalized to arbitrary units, making it impossible to find out whether the single channel limit was indeed exceeded.

- (2) In *Ref.2*, it was shown that larger bounds on *extinction cross section* were possible for non-spherical shapes. The work was mostly centered on quasistatic plasmonic modes featuring strong absorption losses, and therefore in their case extinction could mean the sum of absorption and scattering, *not necessarily scattering*. In particular, the authors find that their upper bound is attained for *strongly absorbing* scatterers subject to the quasistatic approximation.

Furthermore, there was no multipolar analysis involved, and therefore the interesting possibility to boost the dipole scattering cross section was not investigated. We also note that our work is not focused on establishing *bounds* to the cross section for nonspherical shapes, but rather to reveal a novel physical mechanism allowing the formation of super dipole resonances from quasi-BICs as a starting point.

Importantly, we notice that neither of those works mention that symmetry breaking allows to control the Q-factor in addition to the cross section. In the new version of the manuscript, we have shown that this property renders super multipole resonances more resilient to ohmic losses than in conventional superscattering (Figure 5). As explained in the remark 3.1 below,

super multipoles maintain a large scattering cross section but are supported by a mode with lower Q-factor than the original quasi-BIC (please compare Figure 5a-b). Consequently, with increasing ohmic losses, the cross section of a super multipole drops slower (Figure 5c). For a given value of imaginary part of the refractive index, the cross section of a super multipole is always larger than the uncoupled counterpart, rendering non-spherical geometries more practical for applications.

Figure 5. Robustness of super-multipole resonances to Joule losses. Evolution of the total scattering cross section with increasing imaginary refractive index δ for a sphere (a) and an ellipsoid with $r_{\perp} / r_{\parallel} = 0.85$ (b). The two scatterers have an equal volume and real index $n=3.87$. (c) Comparison of superscattering maxima [dashed lines in (a) and (b)] with increasing losses for the quasi-BIC (sphere) and the super-dipole resonance (spheroid).

The two references mentioned by the reviewer are now cited in the main manuscript, where we have also added the following discussion in the section entitled ‘Enhancing scattering by finite objects’:

What happens in the absence of spherical symmetry? Intriguingly, it was shown that significantly larger bounds on total extinction (the sum of scattering and absorption for all multipole channels), could be attained for non-spherical shapes²¹, even for deeply subwavelength plasmonic particles. The enhancement, however, was mostly delivered by strong absorption from such particles. Finite plasmonic nanorods were also numerically investigated²², exhibiting enhanced cross sections for some well-chosen geometrical parameters. Despite these works, focused in plasmonic cavities with large absorption cross sections, there appeared to be no essential differences between a spherical shape and the general case.

Reviewer #2

3) From the result in Fig. 5, the scattering cross section only slightly exceeds the single-channel limit (about 1.5 times).

Our reply

We would like to emphasize that recent works on superscattering in the visible, based on overlapping the electric and magnetic dipole resonances, achieved similar cross section values that we attained with our novel super ED (**Figure 4a** in Lepeshov et.al. *ACS Photonics* **6**, 2126–2132 (2019), **Figure 3a** in Yang, Yi, et al. "Low-loss plasmonic dielectric nanoresonators." *Nano letters* 17.5 (2017): 3238-3245). While validated in the microwave, our design was originally targetted to be implementable with materials available in the visible and/or IR. Furthermore, while we enhance the ED cross section by approximately 1.6 times, the total cross section also has some contributions from the MQ cross section, as discussed in the main text. The **total cross section** (in the absence of absorption loss) **exceeds 2**, as is now shown in the new Figure 1b.

Therefore, the values presented in the original design are comparable and even higher than the earlier literature, and we can firmly claim we reach the superscattering regime.

However, to address the Reviewer's concerns, in the new version of the manuscript, Figure 8b presents the calculations for a new design (a rectangular prism with $n = 3.3$, e.g., GaAs in the near-IR), where we have managed to boost the **magnetic dipole cross section** by almost three times the single channel limit, thus doubling our earlier result.

Figure 8. Super MD resonance produced by dipole-octupole hybridization. (a) Inset: scheme depicting the geometry of the cavity and the incident illumination. The initial cavity is a rectangular prism with unequal sides $L_x = 500$ nm, $L_y = 435$ nm, $L_z = 260$ nm, and refractive index $n=3.3$. L_x is kept constant, while L_y and L_z are progressively tuned by an amount δ . Q-factors of the two involved QNMs as a function of δ . The quasi-BIC appears in the lower branch (blue circle), characterized by an overall high Q-factor, for $\delta = -40$ nm, while the super MD appears at the upper branch (with overall low Q-factor) for a detuning of $\delta = 50$ nm from the initial geometry (red circle). It coincides with a slight enhancement of the Q-factor of the upper branch and a decrease in the lower one. (b) Multipole decomposition of the scattering cross section for the cavity with $\delta = 50$ nm, normalized by the dipole limit. We observe a remarkable enhancement of the MD cross section of more than 2.6 times the conventional limit. The latter is almost equivalent to the cross section of three magnetic resonances of isolated spheres at the same wavelength (see inset). Incidentally, we also observe the accidental overlap of the high-Q branch, radiating as a combination of electric quadrupole (EQ) and MD, with a super ED resonance like the nanorod in the previous section. (c) Comparison between the scattered fields produced by a perfect electric conductor (PEC) cavity (upper panel) and the designed superscatterer (lower panel), both with the same dimensions. The fields are recovered at the wavelength of 935 nm, corresponding to the large MD peak in (b). (d) Near field distribution of the involved QNMs for selected δ (refer to discussion in text).

Reviewer #2

3.1.) I suggest the authors give us more convincing results in the presence of loss.

Our reply

It is important to emphasize that our super ED is implemented with a high-index dielectric scatterer, which generally features small, even negligible ohmic losses. In particular, the experiment was carried out with a ceramic with permittivity $\epsilon = 22$ and loss tangent 0.001. Figure S3 of the Supplementary Information shows that, even in this case, the ED cross section exceeds the single channel limit. This is also confirmed in experiment (Figure 5a).

Similar loss tangents, or even smaller losses are typical for amorphous silicon in the near-IR part of the visible range.

Despite that, we absolutely agree with the Reviewer that a thorough analysis in the presence of losses is necessary. For that reason, Figure 5 now displays a comparison between the effect of losses in the dielectric sphere and the dielectric spheroid.

Figure 5. Robustness of super-multipole resonances to Joule losses. Evolution of the total scattering cross section with increasing imaginary refractive index δ for a sphere (a) and an ellipsoid with $r_{\perp} / r_{\parallel} = 0.85$ (b). The two scatterers have an equal volume and real index $n=3.87$. (c) Comparison of superscattering maxima [dashed lines in (a) and (b)] with increasing losses for the quasi-BIC (sphere) and the super-dipole resonance (spheroid).

As a result of our analysis, we have found that super multipole resonances are more resilient to losses than high-order multipole resonances in a sphere. Indeed, one important drawback of conventional superscattering is the fast degradation of the effect with intrinsic losses. While spectrally overlapping high order multipole resonances strongly maximizes scattering, once ohmic losses are not negligible the scattering cross section drops very rapidly. Consider a dielectric scatterer with refractive index $n_p = n + i\delta$. As we show in the new Supplementary Information S5, large Q-factors imply a rapid drop of the cross section maxima, yielding a slope for small δ of $d\sigma/d\delta \approx -4Q/\omega_0$, where ω_0 is the resonance frequency. Since high-order multipole resonances are associated with large Q-factors, their maximum scattering cross section decreases rapidly with increasing δ . Herein the reason why almost a decade passed since the original proposal until the experimental demonstration of superscattering in Ref. Phys. Rev. Lett. 122, 063901 (2019).

As an example, consider the quasi-BIC resonance for the sphere case, displayed in Figure 5a (indicated by the dashed line). In the lossless scenario, a strong scattering peak can be appreciated, reaching the maximum allowed for the MQ, i.e., $5\lambda^2/2\pi$. Thus, if one is only interested in the overall scattering cross section, there is no apparent need to ‘transform it’ into a super-dipole resonance, since the quasi-BIC already provides a significant scattering enhancement beyond the dipole limit. Moreover, by adding a shell of a different material, one could spectrally overlap the quasi-BIC with, e.g., the electric dipole, to yield a large enhancement. However, there is a caveat: increasing δ of the sphere by only 0.03 results in a drastic drop of the scattering cross section by more than 80%, even below that of a conventional dipole resonance. Therefore, in a practical scenario, high-order multipole resonances are not ideally suited to deliver the desired cross section.

Critically, super-multipole resonances, when formed through the FW mechanism, offer the ability to control the Q-factor. As we now discuss in the first section of the new manuscript, if the QNM is compatible with two scattering channels, the radiation losses increase, but contrarily to common belief, the total scattering cross section is not degraded, and can even increase at a super-multipole resonance. As a result, these novel states are more resilient to intrinsic losses, since the slope $d\sigma/d\delta$ is smaller than the original uncoupled resonances.

Figure 5b illustrates this with the example of the super-ED resonance. The sphere of Figure 5a is deformed into an ellipsoid, and the quasi-BIC evolves into a super-ED with lower Q-factor (dashed line in Figure 5b). To make a fair comparison, the volume of the nanocavity is kept constant. In stark contrast with the quasi-BIC, the drop in the cross section is appreciably smaller, and for $\delta=0.03$ it remains above the single-channel limit.

A comparison between the maximal cross section attained by the quasi-BIC and the super-ED with increasing δ is displayed in Figure 5c. For small δ , we confirm that the quasi-BIC maximum has a much steeper slope as a function of δ , while from the start, the super-ED keeps a much higher cross section. Thus, due to their inherent robustness to losses, super-multipole resonances are better candidates to enhance the scattering cross section at the nanoscale.

We have added the previous discussion and Figure 5 to the end of the section entitled 'subwavelength nanoresonators'.

Reviewer #2

3.2.) Also, I suggest the authors compare the scattering performance with other object, such as a PEC disk with the same geometry.

Our reply

We thank Reviewer #2 for this suggestion. In the new version of the manuscript, we have plotted the scattered fields produced by the optimized rectangular prism and compared them with those of a PEC scatterer of the same dimensions (Figure 8c). It is worth noting that a significant enhancement can be appreciated for the dielectric prism, both in the forward and in the *backward* directions. This constitutes an important difference with respect to conventional superscattering. In the vast majority of designs only forward scattering can be maximized^{14,17,20,24}. Thus, super-multipole resonances offer an attractive strategy to enhance backscattering without the need to sacrifice the overall scattering efficiency, as is the case for the anti-Kerker effect³³.

Figure 8. Super MD resonance produced by dipole-octupole hybridization. (a) Inset: scheme depicting the geometry of the cavity and the incident illumination. The initial cavity is a rectangular prism with unequal sides $L_x = 500$ nm, $L_y = 435$ nm, $L_z = 260$ nm, and refractive index $n=3.3$. L_x is kept constant, while L_y and L_z are progressively tuned by an amount δ . Q-factors of the two involved QNMs as a function of δ . The quasi-BIC appears in the lower branch (blue circle), characterized by an overall high Q-factor, for $\delta = -40$ nm, while the super MD appears at the upper branch (with overall low Q-factor) for a detuning of $\delta = 50$ nm from the initial geometry (red circle). It coincides with a slight enhancement of the Q-factor of the upper branch and a decrease in the lower one. (b) Multipole decomposition of the scattering cross section for the cavity with $\delta = 50$ nm, normalized by the dipole limit. We observe a remarkable enhancement of the MD cross section of more than 2.6 times the conventional limit. The latter is almost equivalent to the cross section of three magnetic resonances of isolated spheres at the same wavelength (see inset). Incidentally, we also observe the accidental overlap of the high-Q branch, radiating as a combination of electric quadrupole (EQ) and MD, with a super ED resonance like the nanorod in the previous section. (c) Comparison between the scattered fields produced by a perfect electric conductor (PEC) cavity (upper panel) and the designed superscatterer (lower panel), both with the same dimensions. The fields are recovered at the wavelength of 935 nm, corresponding to the large MD peak in (b). (d) Near field distribution of the involved QNMs for selected δ (refer to discussion in text).

Reviewer #2

4) The authors stress that “our mechanism is in stark contrast with the conventional way to achieve superscattering”. I find the underlying mechanism is the cross of two modes to create a super dipole mode. In this regard, the working mechanism is similar to some extent.

Our reply

We cannot agree with this comment since the fundamental mechanisms leading to conventional superscattering and our supermultipoles are different. To illustrate it in more detail, we have added Figure S4 (shown below) to the supplementary information. Here, with the example of a Si cylinder, we compare the two effects. Figures S4a and S4c depict the typical situation leading to conventional superscattering. They show, respectively, the resonant frequencies and the scattering cross sections of the electric dipole (ED) and magnetic dipole (MD) QNMs. By varying the height of the cylinder, **the two resonant frequencies cross**, forming an accidental degeneracy. As a result, the total scattering cross section is the sum of the ED and MD contributions and exceeds the single channel limit (Figure S4c). However, the cross section of each multipole is still **bounded by the limit**.

In contrast, Figures S4b,d show the first case studied in the main manuscript. The first noticeable difference is that no crossing takes place between the QNMs u and d (Figure S4b). As explained in detail in the text, this is due to strong coupling. Despite this, the total scattering cross section has even higher values than a conventional superscatterer (Figure S4d). Secondly, at the super dipole resonance, the ED cross section has a dominant contribution to scattering and is no longer bounded by the limit. Namely, it breaks this previously accepted fundamental constraint for subwavelength passive scatterers.

Figure S4. Comparison between the conventional superscattering effect and a super dipole resonance. The numerical results are obtained for a series of Si nanocylinders of radius $r=126$ nm and varying height H . (a) Resonant frequencies of the MD and ED QNMs in the vicinity of an accidental degeneracy. (b) Avoided crossing of the two hybrid QNMs studied in the main text. (c) Multipole decomposition for $H=95$ nm, corresponding to the maximal overlap between the ED and MD QNMs. The multipoles do not exceed the single channel limit, but their sum does. (d) Super dipole resonance, also indicated by the black circle in panel (b). Here, the electric dipole can exceed the single channel limit, and there is no need for crossing of the resonant frequencies

Reviewer #2

5) Please state the practical significance and application of conventional superscattering and symmetry breaking superscattering in detail. It would be better to show **several specific examples**, beyond the theoretical study.

Our reply

We agree with Reviewer #2 that more emphasis needed to be made on key practical differences with respect to conventional superscattering. On the one hand, we stress that our mechanism allows to simultaneously control the **overall scattering and Q-factor** of the involved resonances. The latter is not possible with conventional superscattering.

Supermultipoles are in consequence more resilient to ohmic losses, which will greatly facilitate their implementation into practical devices. On the other hand, our mechanism allows us to **selectively enhance** the cross section of some multipoles on behalf of others. This leads to an improved flexibility to engineer multipolar interference effects. For instance, as shown in Figure 8c, we can boost the backscattering cross section at the same time as forward scattering, without compromising the overall scattering efficiency. Alternatively, the super ED and super MD resonances could be combined to form a ‘super’ Huygens dipole, with important implications for e.g., sensing of chiral molecules.

On the other hand, larger scattering cross sections also lead to enhanced optical forces. The directionality of the latter is essentially determined by the scattering pattern (i.e., the excited multipoles, c.f. Advanced Photonics Research 2.9 (2021): 2100082). While the conventional superscattering mechanism could help boost optical forces, it does not allow control over their direction, since there is no control over the multipolar characteristics of the overlapping resonances. In contrast, super multipoles allow to selectively enhance the contribution of one multipole to scattering, thus allowing to engineer not only the intensity, but also the **directionality** of optical forces. Thus, we pave the way towards a more flexible engineering of optical forces at the nanoscale, which could be relevant for the next generation of all-optical nanomachines.

We have added the following text to the conclusion section “Besides their fundamental interest, such exotic scattering can be employed in biosensing^{34,35} or energy harvesting^{36–38} devices. Strongly scattering dielectric nanoantennas, operating in a selective polarization regime, can replace their plasmonic counterparts as ultra-compact demultiplexers for on-chip circuitry³⁹. Furthermore, the ability to selectively enhance the scattering pattern of a given multipole can unlock new degrees of freedom for optical manipulation^{40,41} .”

Reviewer #2

6) In my understanding, superscattering is a device to help other objects enhance their scattering, just like invisibility cloak that wraps up other objects. I am wondering superscattering cannot realize this goal because it only enhances itself scattering. How to understand this point?

Our reply

Indeed, originally, superscattering was proposed as the counterpart of an invisibility cloak. However, the benefits of an enhanced cross section extend far beyond this first application. A larger scattering cross section implies that more photons can be captured per unit area. This has direct implications for the realization of, e.g., advanced solar cells, see *Journal of Optics* 18, 015903 (2015). The implications go beyond optics. For instance, in acoustics, superscattering objects enable directional sensing at low frequencies, see *Physical Review Applied* 5, 054015 (2016).

Until now, the rational design of devices supporting this effect was generally limited to geometries where the early theory was still applicable (where the single channel limit is still valid). We believe our work is a step forward to implement the superscattering effect in practical settings. We emphasize that, unlike the original proposal, super multipoles offer an efficient control not only over the scattering efficiency, but also over the far field and the resonance linewidth. We envision that our effect could be useful, for instance, to realize efficient waveguide couplers, where an enhanced scattering efficiency and the control over the scattering pattern directionality are required.

REVIEWER COMMENTS

Reviewer #1 (Remarks to the Author):

The authors have revised the manuscript in accordance with the comments. I have no more questions and would like to recommend publishing as present.

Reviewer #2 (Remarks to the Author):

I appreciate the authors for the great efforts they have made. Before making a final decision, I still have several comments that are important for this superscattering work.

1) Foremost, we have to admit a fact that single-channel limit is a strict theoretical upper limit. No one can touch and break this limit in a passive system. The authors emphasize that they break the single-channel limit using an asymmetrical passive structure. Intuitively, this is anti-physical. The only condition to allow it happen is that the definition of single-channel limit in the authors' system is different from previous definition, which is also mentioned by Referee 1 comment 1. If this is the case, I strongly suggest the authors modify the clarification.

2) In the comparison with Nanoscale 6, 9093–9102 (2014) & Phys. Rev. Lett. 112, 113903 (2014), I am not convinced enough about the explanation because they also use nanodisk structure and achieve superscattering.

3) The performance of superscatterer is not very eye-catching, such as 1.6 or 2 times of single-channels. In previous superscattering works, the total scattering strength is larger than 5 times and even more.

4) Another important question is that, since the first propose of superscattering in PRL, 2010, pursuing an interesting and practical-oriented application is always a long-standing dream. Superscatterer is a functional device that makes itself enhance scattering. In practice, we may more want superscatterer can help other object enhance scattering, just like invisible cloak. I want to know how to specifically bring superscattering into applications and help other object enhance scattering.

**Reply to the second report of Reviewer #2
and a summary of the changes made in the revised manuscript**

We are grateful for additional comments of Reviewer #2 on our manuscript NCOMMS-21-51814A. Below, each comment is quoted in *italics*, and it is followed by our response. We have also revised the manuscript accordingly and supply a marked manuscript.

Comment 1:

Foremost, we have to admit a fact that single-channel limit is a strict theoretical upper limit. No one can touch and break this limit in a passive system. The authors emphasize that they break the single-channel limit using an asymmetrical passive structure. Intuitively, this is anti-physical. The only condition to allow it to happen is that the definition of single-channel limit in the authors' system is different from previous definition, which is also mentioned by Referee 1 comment 1. If this is the case, I strongly suggest the authors modify the clarification.

Our Response:

We understand Reviewer's concern regarding the definition of the single-channel limit, since this is a critical aspect of our work. We emphasize that our definition is **identical** to the one used in the seminal work of Fan et.al¹. In their original paper and follow-up², each multipole moment is assumed to be bound by the limit $\sigma_{\text{Max}} = \frac{2\ell+1}{2\pi} \lambda^2$, where ℓ is the angular momentum of the multipole. In particular, $\ell=1$ (dipolar scattering), yields $\sigma_{\text{Max}} = \frac{3}{2\pi} \lambda^2$. The critical difference is that we deal with a much broader class of scatterers that do not have spherical shape. In this case, in the section entitled 'enhancing scattering by finite objects', we show that the conventional bound **no longer applies**.

This is because the so-called 'reflection matrix', which connects incoming and outgoing waves from the scatterer, is no longer necessarily diagonal, as was originally assumed by Fan and coworkers.

In our work, we demonstrate a mechanism to enhance this effect, maximizing the scattering cross section of one multipole in a passive structure, which is in strong contrast to what has been realized so far.

In particular, Figure 1b (attached below), shows that the super-ED resonance boosts the ED cross section by almost double the conventional single-channel limit.

Figure 1. (a) Concept of BIC-inspired superscattering in an isolated resonator. Strong coupling of two modes reshapes both their near fields and scattering patterns as a function of a tuning parameter. The interfering resonances lead to a quasi-BIC state (destructive interference) and induce power redistribution between multipolar scattering channels leading to super-dipole radiation (constructive interference). (b) Super dipole resonance arising in the scattering cross section of a dielectric cylinder with refractive index $n = 3.8$, radius 130 nm and height 180 nm. The scattering cross section of the electric dipole channel is seen to significantly exceed the single-channel limit. This is in contrast with conventional superscattering, where several multipole resonances need to be overlapped.

To further demonstrate to the Reviewer that we are using the correct limit, we have calculated (using rigorous Mie theory) the scattering cross section of the first ED resonances in a lossless sphere with $n=4$ (Figure R1, blue line), and normalized them by the limit used in the main text.

The Reviewer can confirm that the maxima of the resonances in the sphere *is always bound by the single channel limit*. In strong contrast, the optimized ellipsoid from the main text (Figure R1, black line) displays the novel super-ED resonances, where *the maximum significantly exceeds the single-channel limit*.

Figure R1. Lowest order ED resonances of a Si sphere and a Si ellipsoid ($n=4$) with aspect ratio $b/a=0.8$ vs. a/λ . b , a are the semiaxis of the ellipsoid. a is kept fixed at 100 nm. For the sphere, all the resonances are bounded to the single channel limit. However, the optimized ellipsoid can greatly surpass this limit. Moreover, the maximum enhancement can be clearly seen to occur at the super ED resonance described in the main text. Only the ED scattering cross section is plotted.

We have extended the Supporting Information with Figure R1 (Figure S5) and the discussion above, in the new section S11. We have also referred the reader to section S11 in the main manuscript.

Comment 2:

In the comparison with Nanoscale 6, 9093–9102 (2014) & Phys. Rev. Lett. 112, 113903 (2014), I am not convinced enough about the explanation because they also use nanodisk structure and achieve superscattering.

Our Response:

We agree with the Reviewer that a more careful comparison with these previous works is necessary, particularly *Nanoscale 6, 9093–9102 (2014)*. In the latter, the authors considered a Ag nanocylinder embedded in a polymer with refractive index 1.4, and claimed to reach the superscattering regime. We note that all the cross sections in their work are normalized to arbitrary units, making it difficult to determine the scattering enhancement. Figure R2(a) shows our calculations of the scattering cross section using the same parameters (details are given in the caption).

The calculated scattering cross section is only slightly above the single-channel limit (approximately 1.2 times). We attribute the small efficiency to the presence of Ohmic

losses from Ag. Most importantly, the mild enhancement is due to the spectral overlap of the ED and EQ resonances, which are associated to two distinct eigenmodes of the cylinder [Figure R2(b)]. Each multipole, by itself, does not exceed the single-channel limit. Hence, despite investigating a similar geometry, the authors did not observe the same effect that we are describing in our work, where a single multipole exceeds the limit.

Figure R2. Comparison with earlier works that demonstrated enhanced scattering with non-spherical shapes in/ near the visible range. (a)-(b) cylindrical plasmonic nanocavity studied in Ref. ³. Following the latter, the nanoparticle is described by the Drude model $\varepsilon(\omega) = \varepsilon_\infty - \frac{\omega_p^2}{\omega(\omega + i\gamma_c)}$, with the following parameters: $\varepsilon_\infty = 4$, $\omega_p = 1.336 \cdot 10^{16}$ rad/s, $\gamma_c = 1.108 \cdot 10^{14}$ rad/s. The scatterer is embedded in a host environment with refractive index $n_h = 1.4$, hence the single channel limit is given by $\sigma_{\text{Max}} = \frac{3}{2\pi n_h} \lambda^2$.

The diameter of the cylinder is 80 nm, and the height is 120 nm. All corners have been rounded to avoid unphysical field singularities. (a): Multipole decomposition of the scattering cross section. Inset: scheme depicting the illumination setup. The shaded green area indicates the spectral region with maximum scattering enhancement. The enhanced scattering occurs due to the overlap of the ED and EQ contributions, following the conventional mechanism of Fan et al ^{1,2}. Note that **none of the two multipoles exceeds the single channel limit**, and the total cross section is only slightly above it. (b) Upper panels: field distributions of the two involved QNMs. The rightmost panel is the ED QNM, and the leftmost is the EQ QNM. Lower panels: Evolution of the surface charge during half an oscillation period

at the wavelength with maximum scattering enhancement. (c)-(d) Ag ellipsoids investigated in Ref.⁴. (c) Multipole decomposition of the scattering cross section of an ellipsoid composed of Ag (data obtained from Johnson and Christy ⁵), with semi-major axis 10 nm. The scattering cross section is entirely dominated by the ED, and shows values almost **1000 times below the single channel limit**. Inset: scheme of the illumination setup and the scatterer under consideration. (d) Comparison between scattering and absorption efficiencies (in arbitrary units). The absorption cross section is almost 20 times larger than the scattering cross section.

Next, we consider the structures investigated in *Phys. Rev. Lett.* 112, 113903 (2014). The authors considered Ag ellipsoids of extreme subwavelength sizes, in the order of 6% the wavelength, [as depicted in the inset of Figure R2(c)]. We have calculated the scattering cross section of one of such ellipsoids near their localized plasmon resonance, [Figure R2(c)]. The Reviewer can confirm that the scattering cross section is **1000 times below the single channel limit**. Therefore, these ellipsoids are very weak scatterers. Moreover, Figure R2(d) compares the magnitudes of the scattering and absorption efficiencies. Absorption is almost 20 times larger.

To understand these results, we should clarify that the authors in *Phys. Rev. Lett.* 112, 113903 (2014) were not interested in maximizing the scattering cross section of a single particle; their figure of merit was the **extinction per unit volume**. In addition, as we have now verified with full-wave simulations, the response of such particles will always be strongly dominated by absorption, in contrast to our dielectric nanoparticles.

We have extended the Supporting Information with Figure R2 (Figure S6) and the discussion above, in the new section S12. We have also referred the reader to section S12 in the main manuscript.

Comment 3:

The performance of superscatterer is not very eye-catching, such as 1.6 or 2 times of single-channels. In previous superscattering works, the total scattering strength is larger than 5 times and even more.

Our Response:

We are thankful to the Reviewer for insisting in this point!

We would like to remark the fact that our initial structure displays a scattering cross section that is comparable and even larger than recent works on superscattering in the visible -

telecommunication range, based on overlapping the electric and magnetic dipole resonances. To show this, we have reproduced the results of these works in Figure R3.

Figure R3. Comparison with previous exemplary proposals to achieve superscattering in the visible/telecommunication range in realistic designs. (a) Scattering cross section of the core-shell sphere proposed in Ref. ⁶. The Ag data was taken from Palik (as in Ref. ⁶), and the GeTe dispersion was directly taken from Fig. S1 of Ref. ⁶. Geometrical parameters: $R = 300$ nm and $t = R / 4$. Calculations were performed analytically with conventional Mie theory. Inset depicts the geometry and the incident illumination scheme. (b) Scattering cross section of the Si cylinder studied in Ref. ⁷. Inset: geometry and illumination scheme. Geometrical parameters: $H=100$ nm, $D=240$ nm.

The main novelty of our work is the discovery of a new effect to reach the superscattering regime: super-multipoles. These unusual resonances allow to exceed by themselves the single channel limit. As the Reviewer can verify from Figure 1(a), the scattering cross section of the super-ED resonance is comparable to the sum of the conventional ED and MD resonances in Figure R3(b).

Most importantly, this implies that we can **combine** both mechanisms: i.e. supermultipoles and overlapping resonances, to enhance scattering even more. In the new version of the manuscript, we have designed a dielectric nanocylinder to feature both effects simultaneously at the same wavelength (new Figure 9, shown below). This is done by forcing the crossing of the MD and the super-ED resonances [Figure 9(a)]. By doing so, the total cross section reaches 5 times the single-channel limit [Figure 9(b)], which is significantly larger than the previous works in the visible, and larger even than much more complicated designs in the microwave ⁸ and telecommunication range ⁹.

In the new submission, we have expanded the section “Boosting a super-multipole even further” with Figure 9 and two explanatory paragraphs.

Figure 9. Combining the old and the new mechanisms to achieve even more superscattering from subwavelength nanostructures. (a) Calculated resonance wavelengths of the even and odd QNMs for a dielectric nanocylinder with $n = 3.35$ and radius 160 nm. The even QNMs are spectrally close and they couple, resulting in the formation of a quasi-BIC and a super-ED as a function of height (H). Now, however, we designed the nanoparticle such that the super-ED spectrally overlaps with an odd QNM (accidental crossing), following the original strategy by Fan et.al 16. (b) Scattering cross section for $H = 300$ nm. The super-ED boosts the ED cross section by 3 times the single-channel limit. In combination with the MD and higher order multipolar contributions, the total cross section reaches 5 times the limit. (c) x-component of the electric field at $\lambda = 565$ nm. The field can be seen to be strongly distorted by the scatterer. (d) Comparison between the far fields produced by a perfect electric conductor (PEC) cavity (green pattern) and the designed superscatterer (purple pattern), both with the same dimensions. The fields have been normalized to the maximum of the superscatterer.

Comment 4:

Another important question is that, since the first propose of superscattering in PRL, 2010, pursuing an interesting and practical-oriented application is always a long-standing dream. Superscatterer is a functional device that makes itself enhance scattering. In practice, we may more want superscatterer can help other object enhance scattering, just like invisible

cloak. I want to know how to specifically bring superscattering into applications and help other object enhance scattering.

Our Response:

This is indeed an important question. The superscattering effect should be understood to make **a particular object** scatter more photons per unit area. Alternatively, to use the superscatterer to enhance scattering of other objects besides itself, one could conceive metasurface coatings, similar to the ones proposed in Ref. ¹⁰. However, we emphasize that the superscattering effect has a much broader application scope. We highlight, for example, advanced solar cells¹¹, acoustic directional sensing¹² and sound mitigation¹³. Scattering enhancement is also of paramount importance in antenna applications¹⁴. A very recent breakthrough, published during the review process of our work, has allowed for the first time to employ this effect for the realization of invisible electromagnetic gateways¹⁵.

Moreover, the first experimental confirmation of the superscattering effect ¹⁶ was published only three years ago. In consequence, we anticipate several new applications in electromagnetism to appear in the next years. Importantly, its practical implementation has been hindered by the strong absorption losses of previous designs, as has been recognized in contemporary works ¹⁶. Another important drawback is the technical complexity of prior proposals ^{10,17,18}. This is particularly critical in optics applications. Due to our all-dielectric implementation, and the robustness of super multipoles to ohmic losses, our design overcomes the previous bottlenecks, and paves the way to utilizing superscattering in the visible range.

As an example of application, we have demonstrated that our superscatterer can be used to hide several objects (or one, but geometrically larger than the superscatterer) from the radiation, e.g., from the scattering force of an incident beam (new Figure 10). Figure 10(a) shows the main idea: The scatterers can be hidden in the large 'shadow' of the superscatterer, which significantly reduces the scattering force experienced by them [Figure 10(b)]. Figure 10(c) shows the calculated ratio of optical force with and without the superscatterer. The Reviewer can appreciate that the scattering force is strongly reduced in a spatial extension much larger than the superscatterer geometrical cross section (diameter 320 nm).

Reducing the influence of scattering forces is essential for efficient optical traps for experiments in atom cooling or biology^{19,20}. Moreover, the effect could also be used in acoustics, e.g., to protect selected volumes from noise utilizing comparably small objects.

Figure 10. Protecting other scatterers from radiation. (a) Conceptual scheme. The superscatterer interacts with photons in a much larger area than itself. As a result, the Poynting vector field lines (purple arrows) are deflected, and the superscatterer leaves a large ‘shadow’, much larger than its diameter¹. The scatterers placed within that shadow (grey shapes) are ‘protected’ from the radiation pressure (red arrows) of the incident beam. (b) Electric field norm in the vicinity of the superscatterer, and calculated radiation pressure experienced by dipolar particles (black arrows). The radiation pressure is significantly decreased within the shadow. (c) Ratio between the radiation pressure with and without the superscatterer, experienced by dipolar particles positioned at a distance $z = -1200$ nm from the superscatterer. The origin of coordinates is at the position of the superscatterer. Inset: artistic view of the superscatterer.

In the new submission, we have added the section entitled ‘Shielding nanoparticles from scattering forces’, including Figure 10 and a detailed discussion.

References

1. Ruan, Z. & Fan, S. Superscattering of light from subwavelength nanostructures. *Phys Rev Lett* **105**, 1–4 (2010).

2. Ruan, Z. & Fan, S. Design of subwavelength superscattering nanospheres. *Appl Phys Lett* **98**, 43101 (2011).
3. Wan, W., Zheng, W., Chen, Y. & Liu, Z. From Fano-like interference to superscattering with a single metallic nanodisk. *Nanoscale* **6**, 9093–9102 (2014).
4. Miller, O. D. *et al.* Fundamental limits to extinction by metallic nanoparticles. *Phys Rev Lett* **112**, 1–5 (2013).
5. Johnson, P. B. & Christy, R. W. Optical constants of the noble metals. *Phys Rev B* **6**, 4370–4379 (1972).
6. Lepeshov, S., Krasnok, A. & Alù, A. Nonscattering-to-Superscattering Switch with Phase-Change Materials. *ACS Photonics* **6**, 2126–2132 (2019).
7. Yang, Y., Miller, O. D., Christensen, T., Joannopoulos, J. D. & Soljačić, M. Low-Loss Plasmonic Dielectric Nanoresonators. *Nano Lett* **17**, 3238–3245 (2017).
8. Mirzaei, A., Miroshnichenko, A. E., Shadrivov, I. v. & Kivshar, Y. S. Superscattering of light optimized by a genetic algorithm. *Appl Phys Lett* **105**, (2014).
9. Qian, C. *et al.* Multifrequency Superscattering from Subwavelength Hyperbolic Structures. *ACS Photonics* **5**, 1506–1511 (2018).
10. Qian, C. *et al.* Breaking the fundamental scattering limit with gain metasurfaces. *Nat Commun* **13**, 4383 (2022).
11. Das, S., Kundu, A., Saha, H. & Datta, S. K. Wide angle light collection with ultralow reflection and super scattering by silicon micro-nanostructures for thin crystalline silicon solar cell applications. *Journal of Optics (United Kingdom)* **18**, (2015).
12. Zhu, X., Liang, B., Kan, W., Peng, Y. & Cheng, J. Deep-Subwavelength-Scale Directional Sensing Based on Highly Localized Dipolar Mie Resonances. *Phys Rev Appl* **5**, (2016).
13. Bai, Y., Wang, X., Luo, X. & Huang, Z. Acoustic superscatterer enables remote mitigation of underwater source radiation. *J Appl Phys* **131**, (2022).
14. Balanis, C.A. *Antenna Theory Analysis and Design.pdf*. Wiley & Sons Inc (1997).
15. Ye, K. P., Pei, W. J., Sa, Z. H., Chen, H. & Wu, R. X. Invisible Gateway by Superscattering Effect of Metamaterials. *Phys Rev Lett* **126**, (2021).
16. Qian, C. *et al.* Experimental Observation of Superscattering. *Phys Rev Lett* **122**, 063901 (2019).
17. Cheng, L. *et al.* Superscattering, Superabsorption, and Nonreciprocity in Nonlinear Antennas. *ACS Photonics* **8**, 585–591 (2021).
18. Wang, C. *et al.* *Superscattering of Light in Refractive-Index Near-Zero Environments*. *Progress In Electromagnetics Research* vol. 168 (2020).
19. Jones, P. H., Maragò, O. M. & Volpe, G. *Optical tweezers: Principles and applications*. (Cambridge University Press, 2015).
20. Juan, M. L., Righini, M. & Quidant, R. Plasmon nano-optical tweezers. *Nat Photonics* **5**, 349 (2011).

REVIEWERS' COMMENTS

Reviewer #2 (Remarks to the Author):

We appreciate the efforts the authors have made, and recommend the acceptance of this work.

Reply to the third report of Reviewer #2

Comment:

We appreciate the efforts the authors have made, and recommend the acceptance of this work.

Our Response:

We thank the Referee for his careful reading of the manuscript!